# Genome editing in primary cells and in vivo using viral-derived Nanoblades loaded with Cas9-sgRNA ribonucleoproteins

Philippe E. Mangeot[1], Valérie Risson[2], Floriane Fusil[1], Aline Marnef[3], Emilie Laurent[1], Juliana Blin [1], Virginie Mournetas[4], Emmanuelle Massouridès [4], Thibault J.M. Sohier [1], Antoine Corbin[1], Fabien Aubé[5], Marie Teixeira[6], Christian Pinset[4], Laurent Schaeffer[2], Gaëlle Legube[3], François-Loïc Cosset[1], Els Verhoeyen[1,7], Théophile Ohlmann[1] & Emiliano P. Ricci [1,5]

Programmable nucleases have enabled rapid and accessible genome engineering in eukaryotic cells and living organisms. However, their delivery into target cells can be technically challenging when working with primary cells or in vivo. Here, we use engineered murine leukemia virus-like particles loaded with Cas9-sgRNA ribonucleoproteins (Nanoblades) to induce efficient genome-editing in cell lines and primary cells including human induced pluripotent stem cells, human hematopoietic stem cells and mouse bone-marrow cells. Transgene-free Nanoblades are also capable of in vivo genome-editing in mouse embryos and in the liver of injected mice. Nanoblades can be complexed with donor DNA for "all-in-one" homology-directed repair or programmed with modified Cas9 variants to mediate transcriptional up-regulation of target genes. Nanoblades preparation process is simple, relatively inexpensive and can be easily implemented in any laboratory equipped for cellular biology.

[1] CIRI, Centre International de Recherche en Infectiologie Univ Lyon, Inserm, U1111, Université Claude Bernard Lyon 1, CNRS, UMR5308, ENS de Lyon, F-69007 Lyon, France. [2] Institut NeuroMyoGène, CNRS 5310, INSERM U121, Université Lyon 1, Faculté de Médecine Lyon Est, Lyon 69008, France. [3] LBCMCP, Centre de Biologie Intégrative (CBI), CNRS, Université de Toulouse, UT3, 118 Route de Narbonne, 31062 Toulouse, France. [4] I-STEM/CECS, Inserm, UMR861 28 rue Henri Desbruères, 91100 Corbeil Essonnes, France. [5] LBMC, Laboratoire de Biologie et Modélisation de la Cellule Univ Lyon, ENS de Lyon, Université Claude Bernard Lyon 1, CNRS, UMR 5239, INSERM, U1210, Lyon 69007, France. [6] SFR BioSciences, Plateau de Biologie Expérimentale de la Souris (AniRA-PBES), Ecole Normale Supérieure de Lyon, Université Lyon1, CNRS UMS3444 INSERM US8, 69007 Lyon, France. [7] Present address: CIRI, Université Côte d'Azur, INSERM, C3M, 06204 Nice, France. These authors contributed equally: Valérie Risson, Floriane Fusil, Aline Marnef. Correspondence and requests for materials should be addressed to P.E.M. (email: philippe.mangeot@inserm.fr) or to E.P.R. (email: emiliano.ricci@ens-lyon.org)

Targeted genome editing tools, such as meganucleases (MGN), zinc-finger nucleases (ZFN), transcription activator-like effector nucleases (TALENs) and more recently the clustered regularly interspaced short palindromic repeats (CRISPR) have revolutionized most biomedical research fields. Such tools allow to precisely edit the genome of eukaryotic cells by inducing double-stranded DNA (dsDNA) breaks at specific loci. Relying on the cell endogenous repair pathways, dsDNA breaks can then be repaired by non-homologous end-joining (NHEJ) or homology-directed repair (HDR) allowing the removal or insertion of new genetic information at a desired locus.

Among the above-mentioned tools, CRISPR-Cas9 is currently the most simple and versatile method for genome engineering. Indeed, in the two-component system, the bacterial-derived nuclease Cas9 (for CRISPR-associated protein 9) associates with a single-guide RNA (sgRNA) to target a complementary DNA sequence and induce a dsDNA break[1]. Therefore, by the simple modification of the sgRNA sequence, users can specify the genomic locus to be targeted. Consistent with the great promises of CRISPR-Cas9 for genome engineering and gene therapy, considerable efforts have been made in developing efficient tools to deliver the Cas9 and the sgRNA into target cells ex vivo either by transfection of plasmids coding for the nucleases, transduction with viral-derived vectors coding for the nucleases or by direct injection or electroporation of Cas9-sgRNA complexes into cells.

Here, we have designed Nanoblades, a protein-delivery vector based on friend murine leukemia virus (MLV) that allows the transfer of Cas9-sgRNA ribonucleoproteins (RNPs) to cell lines and primary cells in vitro and in vivo. Nanoblades deliver the ribonucleoprotein cargo in a transient and rapid manner without delivering a transgene and can mediate knock-in in cell lines when complexed with a repair template. Nanoblades can also be programmed with modified Cas9 proteins to mediate transient transcriptional activation of targeted genes.

## Results

**Cas9-sgRNA RNP delivery through MLV virus-like particles (VLPs).** Assembly of retroviral particles relies on the viral structural Gag polyprotein, which multimerizes at the cell membrane and is sufficient, when expressed in cultured cells, to induce release of VLPs into the cell supernatant[2]. When Gag is coexpressed together with a fusogenic viral envelope, pseudo-typed VLPs are produced that lack a viral genome but still retain their capacity to fuse with target cells and deliver the Gag protein`into their cytoplasm. As previously investigated[3,4], we took advantage of the structural role of Gag and designed an expression vector coding for the MLV Gag polyprotein fused, at its C-terminal end, to a flag-tagged version of *Streptococcus pyogenes* Cas9 protein (Gag::Cas9, Fig. 1a). The two fused proteins are separated by a proteolytic site which can be cleaved by the MLV protease to release the Flag-tagged Cas9 (Fig. 1a). By cotransfecting HEK-293T cells with plasmids coding for Gag::Cas9, Gag-Pro-Pol, a sgRNA, and viral envelopes, fusogenic VLPs are produced and released in the culture medium (herein described as Nanoblades). Biochemical and imaging analysis of purified particles (Supplementary Figure 1a, 1b, 1c and 1d) indicates that Nanoblades (150 nm) are slightly larger than wild-type MLV (Supplementary Figure 1b) but sediment at a density of 1.17 g/ml (Supplementary Figure 1c) as described for MLV VLPs[5]. As detected by western blot, Northern blot, mass-spectrometry, and deep-sequencing, Nanoblades contain the Cas9 protein and sgRNA (Supplementary Figure 1 and 2 and Supplementary Data 1). In addition to Gag, Cas9 and envelope proteins,

mass-spectrometry analysis of Nanoblades identified several cellular proteins, mostly membrane-associated proteins (Supplementary Figure 2a and Supplementary Data 1). Interestingly, the packaging of sgRNA depends on the presence of the Gag::Cas9 fusion protein, since Nanoblades produced from cells that only express the Gag protein fail to incorporate detectable amounts of sgRNA (Supplementary Figure 1d). Furthermore, Cas9-dependent loading of the sgRNA within Nanoblades is not limited by the efficiency of the interaction between the Cas9 and the sgRNA, since expressing an optimized version of the sgRNA that improves binding to Cas9[6] does not appear to increase sgRNA levels within purified VLPs (Supplementary Figure 1d see sgRNA(F+E)).

To assess for Cas9-sgRNA RNP delivery efficiency in target cells and induction of genomic dsDNA breaks, we designed Nanoblades with a sgRNA targeting the 45S rDNA loci. Human 45S rDNA genes are present in hundreds of tandem repeats across five autosomes, locate in the nucleolus and are transcribed exclusively by RNA polymerase (Pol) I[7]. Using immunofluorescence microscopy, it is therefore possible to follow the occurrence of dsDNA breaks at rDNA loci with single-cell resolution by monitoring the nucleolus using the nucleolar marker RNA Pol I and the well-established dsDNA break-marker, histone variant γ-H2AX[8], that localizes at the nucleolar periphery after dsDNA break induction within rDNA[9]. U2OS (osteosarcoma cell line) cells transduced for 24 h with Nanoblades programmed with a sgRNA targeting rDNA display the typical γ-H2AX distribution at the nucleolar periphery with RNA Pol I, indicative of rDNA breaks, whilst cells transduced with Nanoblades with control sgRNAs do not (Fig. 1b, top panel). Interestingly, this distribution of γ-H2AX at the nucleolar periphery can be observed as early as 4 h after transduction in 60% of cells with a maximum effect observed at 16 h after transduction, where almost 100% of observed cells display this γ-H2AX distribution (Fig. 1b, bottom panel and quantification below). In comparison, only 60% of cells transfected with a plasmid coding for Cas9 and the sgRNA display the perinucleolar γ-H2AX/RNA Pol I localization 24 h after transfection. Similar results were obtained in human primary fibroblasts with more than 85% cells displaying this distribution after 16 h (Supplementary Figure 1e). These results suggest that Nanoblade-mediated delivery of the Cas9-sgRNA RNP is both efficient and rapid in cell lines and primary human cells. To further confirm these results, we designed and dosed Nanoblades (by ELISA assay using anti-Cas9 antibodies) programmed with a sgRNA widely used in the literature[10] that targets the human *EMX1* gene to induce dsDNA cleavage at a single locus. HEK-293T cells were then transduced with increasing amounts of Nanoblades and gene editing was measured from the bulk population 48 h after transduction (Fig. 1c). Under these conditions, we observed a dose-dependent effect of Nanoblades ranging from 35% of *EMX1* (at 4 pmol of Cas9) editing to 77% of editing at the highest dose (20 pmol) of Cas9 (Fig. 1c).

Because Nanoblades carry cellular proteins from producer cells in addition to Cas9 (Supplementary Data 1), we tested whether these proteins could also be delivered to recipient cells. For this, we over-expressed the firefly luciferase in producer cells and collected Nanoblades targeting *EMX1* from the supernatant. Luciferase-loaded Nanoblades were then used to transduce HEK293T cells for 24 h. Cells were then washed twice in PBS and incubated in fresh medium for 4, 8, 24, and 48 h. Luciferase activity was measured at each time point, as well as in input Nanoblades (Supplementary Figure 2c). As observed, we could detect a mild luciferase signal (4–6% of input) at 4 and 8 h upon transduction. However, the signal rapidly faded at 24 h (2% of input) and

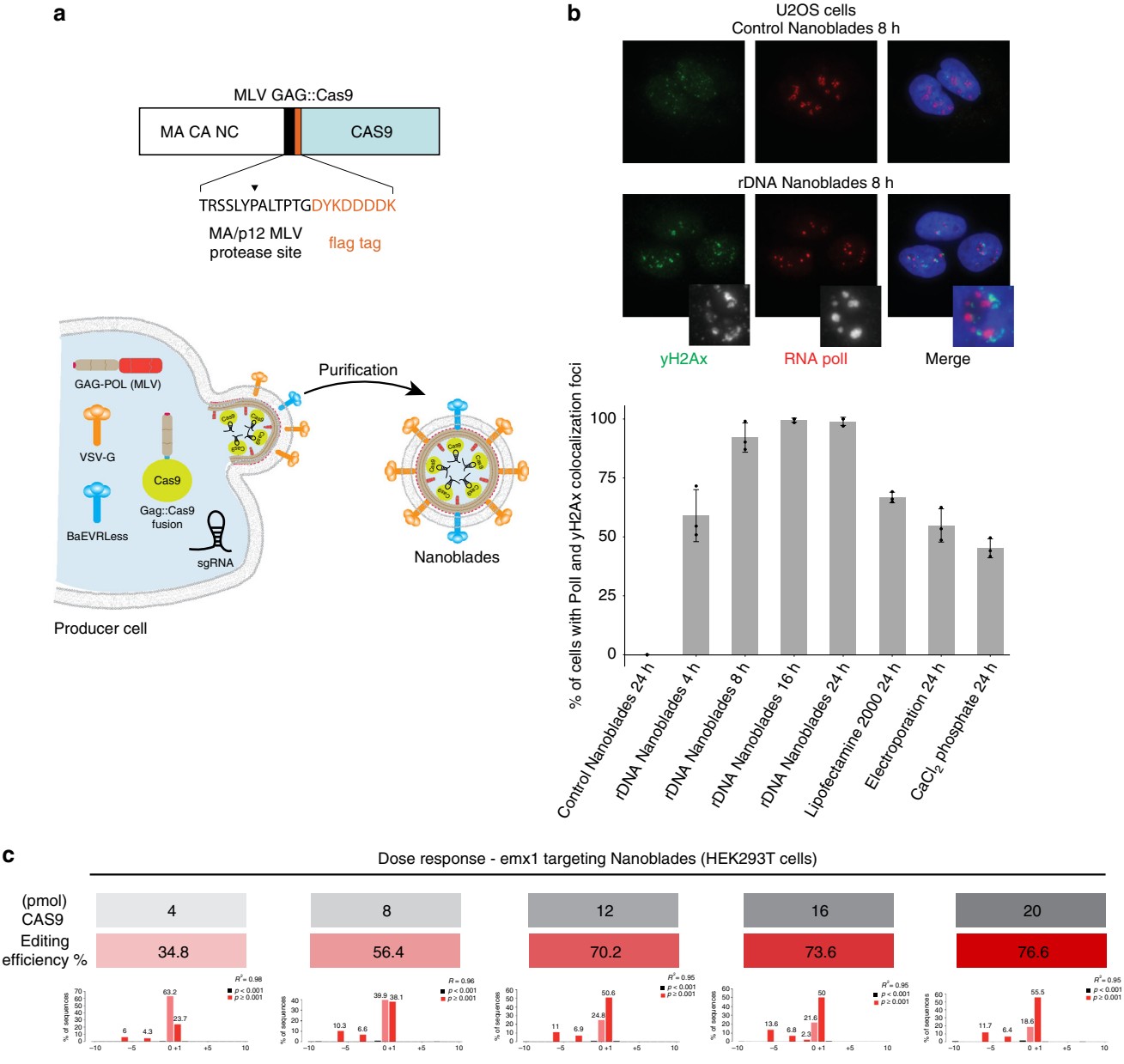

**Fig. 1** Nanoblade-mediated genome editing. **a** Scheme describing the MLV Gag::Cas9 fusion and the Nanoblade production protocol based on the transfection of HEK-293T cells by plasmids coding for Gag-Pol, Gag::Cas9, VSV-G, BaEVRLess, and the sgRNA. **b** Top panel, immunofluorescence analysis of γ-H2AX (green), RNA polI (red) in U2OS cells 8 h after being transduced with control Nanoblades or with Nanoblades targeting ribosomal DNA genes. Bottom panel, quantification of γ-H2AX and RNA polI colocalization foci in U2OS cells at different times after Nanoblades transduction or after classical DNA transfection methods ($n = 3$, error bars correspond to standard deviation). **c** Dose response of Nanoblades. HEK-293T cells were transduced with increasing amounts of Nanoblades targeting human *EMX1* ($n = 1$ displayed). The exact amount of Cas9 used for transduction was measured by dot blot (in gray). Genome editing was assessed by Sanger sequencing and Tide analysis (in red)

was almost undetectable at 48 h (Supplementary Figure 2c). In addition to the ectopically expressed firefly luciferase, we also investigated transmission of the CD81 cell-surface protein, which is highly expressed in HEK293T producer cells and is present in Nanoblades as revealed by mass spectrometry (Supplementary Data 1). HepG2 cells, a hepatic cell line that lacks CD81 expression[11], were transduced for 24 h with Nanoblades targeting *EMX1* and then washed twice with PBS before monitoring CD81 residual signal immediately after the washes or 8 and 48 h after incubation with fresh medium (Supplementary Figure 2d). As observed, even though CD81 was very abundant at the cell surface of producer cells

and completely absent in recipient cells (Supplementary Figure 2d, left and middle panels), we could only detect a mild CD81 signal immediately after transduction (see Supplementary Figure 2d, right panel). Later time points (8 and 48 h) did not show any specific CD81 labeling in recipient HepG2 cells. The impact of cellular proteins delivered by Nanoblades into recipient cells appears therefore limited and restricted to a short time frame.

Taken together, our results indicate that Nanoblades can be efficiently used to mediate genome editing in a rapid and dose-dependent manner with limited impact on the proteome of target cells.

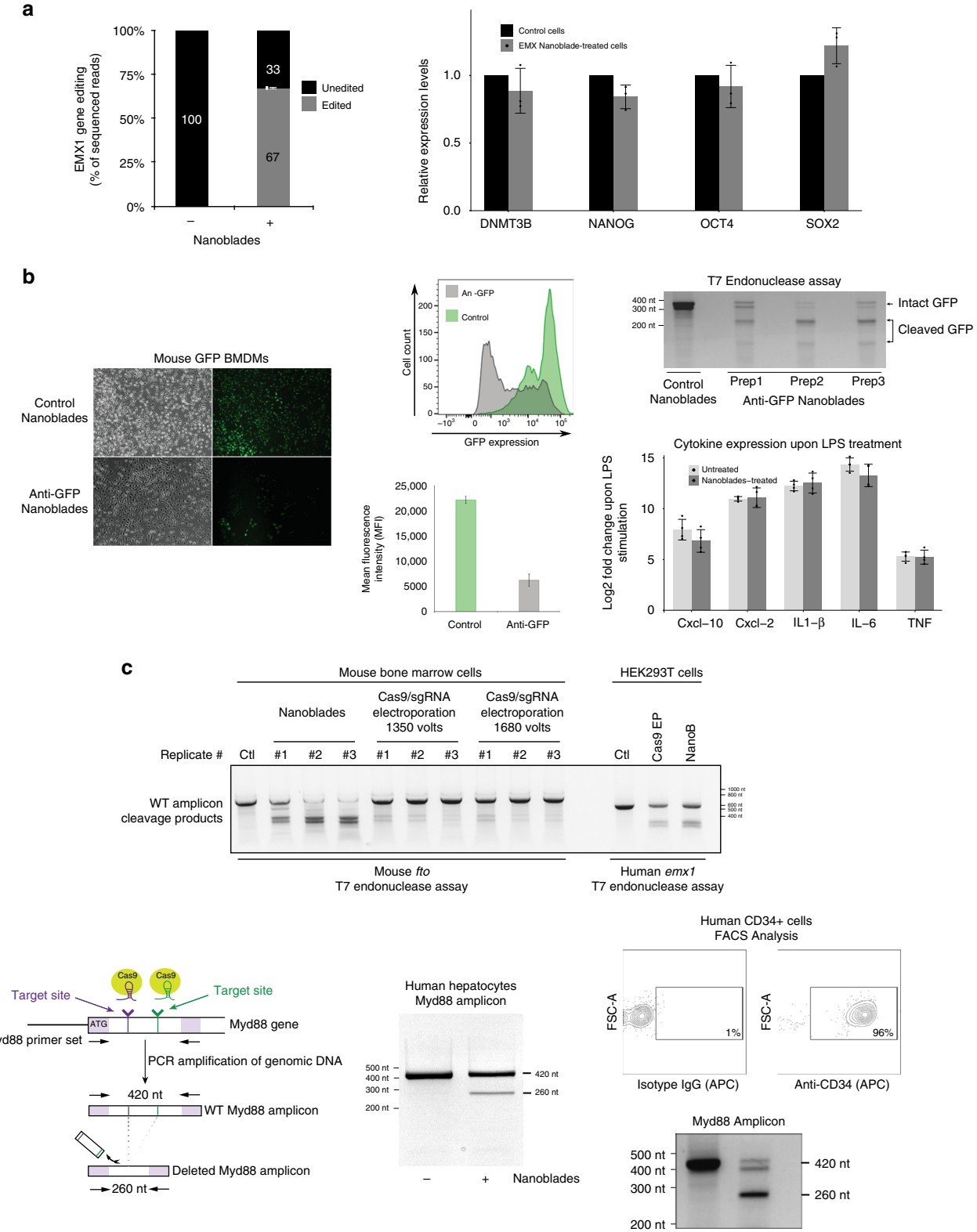

**Nanoblades-mediated genome editing in primary cells**. Genome editing in primary cells and patient-derived pluripotent cells represents a major interest both for basic science and therapeutical applications. However, primary cells are often refractory to DNA transfection and other gene delivery methods. Because Nanoblades were capable of efficient delivery of functional Cas9-sgRNA RNPs into primary fibroblasts, we tested whether they were effective in other primary cells for genome editing. To this aim, Nanoblades targeting *EMX1* were used to transduce human-induced pluripotent stem cells (hiPSCs). Genome editing at the *EMX1* locus was assessed in the bulk cellular population 48 h after transduction by deep-sequencing of the *EMX1* locus (Fig. 2a, left panel). As observed, Nanoblades were capable of mediating 67% genome editing at the *EMX1* locus in hiPSCs. Notably, hiPSCs

**Fig. 2** Genome editing in primary cells transduced with Nanoblades. **a** Left panel, editing efficiency at the *EMX1* locus (measured by high-throughput sequencing on the Illumina Miseq platform) of human-induced pluripotent stem cells (hiPSCs) transduced with Nanoblades targeting human *EMX1* ($n = 3$). Right panel, expression of pluripotency markers measured by qPCR in control cells and cells transduced with Nanoblades targeting *EMX1* ($n = 3$). **b** Left and middle panels, fluorescence microscopy and FACS analysis of GFP expressing BMDMs transduced at the bone marrow stage (day 0 after bone marrow collection) with control Nanoblades or Nanoblades targeting the *GFP*-coding sequence ($n = 3$). Right top panel, T7 endonuclease assay against the GFP sequence from Nanoblades-treated BMDMs. Right bottom panel, cytokine expression levels (measured by qPCR) in untreated or Nanoblade-treated cells upon LPS stimulation ($n = 4$). **c** T7 endonuclease assay against mouse *Fto* or human *EMX1* genomic sequences amplified by PCR from primary mouse bone marrow cells transduced with Nanoblades or electroporated with recombinant Cas9-sgRNA RNPs. For bone marrow cells, two electroporation settings were tested. Lanes numbered #1–#3 correspond to biological replicates. Editing efficiencies were calculated by TIDE[13] analysis of the Sanger sequencing electropherograms for each PCR amplicon **d** Left panel, excision of a 160 bp DNA fragment of *MYD88* using Nanoblades. Middle panel PCR results obtained in human primary hepatocytes transduced with Nanoblades. Right-panel (top), FACS analysis of CD34+ cells purified from human cord-blood. Bottom, genome editing at the *MYD88* locus assessed by PCR in untreated and Nanoblades-treated CD34+ cells. Error bars in all figures correspond to standard deviation

treated with *EMX1* Nanoblades maintained constant levels of pluripotency markers compared to control cells (Fig. 2a, right panel) thus indicating that their multipotent status did not appear to be affected.

Similarly to hiPSCs, mouse bone marrow (BM) cells can be collected and differentiated in vitro into various hematopoietic cell types, such as macrophages (bone marrow-derived macrophages or BMDMs) and dendritic cells. Efficient genome editing of specific genes in BM cells would therefore allow for the corresponding pre-existing protein to be degraded during differentiation and obtain a functional knockout. To test this hypothesis, BM cells obtained from GFP transgenic mice[12] were transduced with Nanoblades programmed with a sgRNA targeting the *GFP* coding sequence. 6 h after transduction, cells were washed and incubated in presence of macrophage colony-stimulating factor (MCSF) for 1 week. After this, cells were collected to monitor GFP levels by fluorescence microscopy, FACS and genome editing by T7 endonuclease assay (Fig. 2b). We consistently obtained close to 75% reduction of GFP expression as measured by FACS analysis and around 60–65% genome editing at the *GFP* locus as measured by T7 endonuclease assays (Fig. 2b). Importantly, genome editing through Nanoblades did not affect the capacity of BMDMs to respond to LPS as their cytokine expression remains identical to that of untreated control cells (Fig. 2b bottom right panel). Nanoblades can therefore be used to inactivate genes in BM cells and study their function in differentiated cells. To further complement these results, we compared the efficiency of Nanoblades to that of recombinant Cas9-sgRNA RNP electroporation in targeting an endogenous gene in primary mouse BM cells. For this, Nanoblades or Cas9-sgRNA RNPs programmed to target the *Fto* gene were used, respectively, to transduce or electroporate primary BM cells freshly extracted from mice. As a control, Nanoblades or Cas9-sgRNA RNPs programmed to target human *EMX1* were also tested in HEK293T cells. In both cases, the efficiency of genome editing was assessed 24 h after transduction or electroporation. As observed (Fig. 2c), both Nanoblades and Cas9-sgRNA electroporation mediate efficient genome editing in HEK293T at 71% (Nanoblades) and 44% (Electroporation) of editing efficiency at the *EMX1* locus. Interestingly, in primary BM cells, while Nanoblades achieve highly efficient genome editing of the *Fto* locus (up to 76% as measured by TIDE[13] analysis), Cas9 electroporation was much less efficient at both conditions that we tested (1350 and 1680 V) yielding a mild but visible signal in the T7 endonuclease assay which was nevertheless below the detection limit for TIDE analysis. Interestingly both protocols (Nanoblades and protein electroporation) did not have an important impact on cell viability 24 h after Cas9 delivery (Supplementary Figure 2e).

Nanoblades efficiency was also investigated in human cells that represent a major interest in research and gene therapy like human primary hepatocytes and human hematopoietic stem cells (HSCs) that both have the capacity to colonize and regenerate fully functional tissues. For both these cell types, Nanoblades programmed with two sgRNAs targeting the human *Myd88* gene were prepared and achieved significant cleavage efficiencies, as revealed by flanking PCR assays (Fig. 2d). Interestingly, HSCs are difficult to transduce with classic VSV-G pseudotyped lentiviral vectors (LVs) because they lack the LDL receptor[14], a limitation that can be alleviated by the use of the baboon retroviral envelope glycoprotein (BaEV)[15]. This prompted us to equip Nanoblades with both BaEV and VSV G-envelopes for these cells and finally in all our study as the combination of both envelopes improved Cas9 delivery in most cells (Supplementary Figure 6a and b). As observed, Nanoblades were also able to induce genome editing in these cells (50% genome editing based on T7 endonuclease assay, Fig. 2d) thus expanding the catalog of primary cells that can be edited using Nanoblades.

Taken together, our results indicate that Nanoblades are an efficient delivery system to induce rapid and effective genome editing in murine and human primary cells of high therapeutic value that are notoriously difficult to transfect.

**"All-in-one" Nanoblades for homology directed repair.** Precise insertion of genetic material (also known as Knock-in) using CRISPR-Cas9 can be achieved through HDR. This occurs when a donor DNA template with sequence homology to the region surrounding the targeted genomic locus is provided to cells together with the Cas9-sgRNA RNP. Based on a previous finding showing that retroviral-particles can be complexed with DNA in the presence of polybrene to allow for virus-dependent DNA transfection[16], we tested whether Nanoblades could be directly complexed with a DNA template to mediate HDR in target cells. To test this approach, Nanoblades programmed to target a locus close to the AUG start codon of the human *DDX3* gene were complexed to a single-stranded DNA oligomer bearing the FLAG-tag sequence flanked with 46 nucleotide (nt) homology arms corresponding to the region surrounding the start-codon of *DDX3* (Fig. 3a, left panel). HEK293T were transduced with these "All-in-one" Nanoblades and passed 6 times before assessing HDR efficiency in the bulk cellular population both by PCR and by Flag-immunoprecipitation followed by western-blotting (using a DDX3 and FLAG-antibody). As observed (Fig. 3a, right panel), cells transduced with "All-in-one" Nanoblades showed incorporation of the FLAG-tag at the *DDX3* locus both genetically and at the level of protein expression (Fig. 3a right panel, see Flag-IP elution and Genotyping panels). In parallel, single-cell

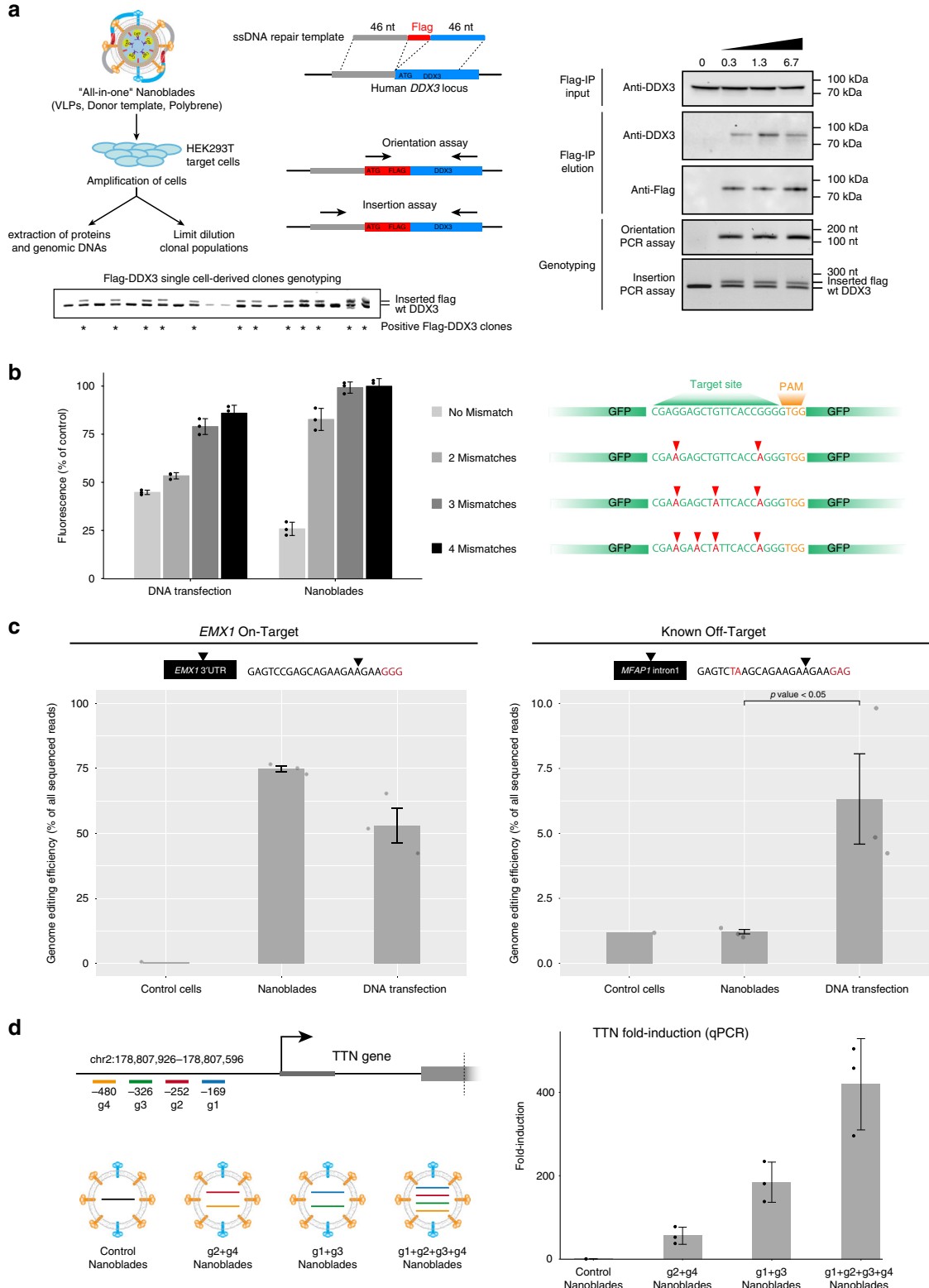

clones were derived from the Flag-DDX3 bulk population and tested for Flag incorporation by PCR. As shown (Fig. 3a left bottom panel), 12 out of 20 isolated clones displayed incorporation of the Flag-sequence at the DDX3 locus thus suggesting a knock-in efficiency of more than 50% of cells using "all-in-one" Nanoblades.

Knock-in assisted by "All-in-one" Nanoblades was also obtained at the AAVS1 locus which has been described as a safe harbor for transgene insertion[17]. For this we designed a dsDNA template of 4 kb bearing the puromycin resistance gene with homology arms to the AAVS1 locus. After transduction of HEK-293T cells with Nanoblades complexed with this template using polybrene, single-cell-derived clones were selected with puromycin. Out of $1 \times 10^5$ transduced cells, we obtained 47 puromycin-resistant clones (Supplementary Figure 3b, c and d). A PCR-assay revealed that 42 out of 47 puromycin-resistant clones tested had

**Fig. 3** "All-in-one" Nanoblades for knock-in experiments and assessment of Nanoblades off-target activity. **a** Left panel, Nanoblades targeting human *DDX3* close to its start codon were complexed with a donor ssDNA bearing homology arms to the targeted locus and a Flag-tag sequence in the presence of polybrene. HEK293T cells were then transduced with these "All-in-one" Nanoblades. After cell amplification, a fraction of cells were collected to extract genomic DNA and total proteins while the remaining cells were cultured to obtain single-cell clonal populations. Right panel, insertion of the Flag-tag in HEK-293T cells transduced with "all-in-one" Nanoblades complexed with increasing amounts of donor ssDNA was assessed by Flag-immunoprecipitation followed by western-blot using anti-flag or anti-DDX3 antibodies in the input and Flag-immunoprecipitation elution fractions. Flag insertion was also assessed by PCR using a forward primer in the flag-sequence and a reverse primer in the DDX3 locus (Orientation PCR assay) or using primers flanking the Flag sequence (Insertion PCR assay). Bottom panel, Flag-insertion in 20 different single-cell-derived clones was assessed by PCR using primers flanking the Flag-sequence. **b** Left panel, off-target monitoring in immortalized mouse macrophages stably expressing *GFP* transgenes bearing silent mutations in the region targeted by the sgRNA. Right panel, cells were transfected with plasmids coding for Cas9 and the sgRNA or transduced with Nanoblades. GFP expression was measured by FACS 72 h after transfection/transduction (*n* = 3). **c** Left and right panels, gene-editing at the *EMX1* on-target site and the *MFAP1* intronic off-target site measured by high-throughput sequencing in untreated cells (control cells) and cells transduced with *EMX1* Nanoblades (Nanoblades) or transfected with plasmids coding for Cas9 and the EMX1 sgRNA (DNA transfection) (*n* = 3). Statistical significance of the Nanoblades and DNA transfection comparison at the on-target site was computed using a two-tail Student test. **d** Left panel, position of sgRNAs targeting the promoter of *TTN* and VLPs with different combination of sgRNAs produced for the experiment. Right-panel, TTN mRNA expression levels (normalized to Control) as measured by qPCR in MCF7 transduced with VLPs (*n* = 3). Error bars in all figures correspond to standard deviation

the puromycin cassette inserted at the AAVS1 locus (Supplementary Figure 3d).

Taken together, our results show that Nanoblades can be used for the precise insertion of genetic material through HDR both with ssDNA and dsDNA donor DNA template and no requirement for any transfection reagent.

**Nanoblades confer low off-target genome-editing.** A major concern related to the use of CRISPR/Cas9-mediated gene editing are the potential off-target effects that can occur at genomic loci that are similar in sequence to the original target. Interestingly, several reports have shown that transient delivery of the Cas9-sgRNA complex by injection or RNP transfection generally leads to reduced off-target effects as compared to constitutive expression of Cas9 and sgRNA from DNA transfection experiments[18]. Since Nanoblades deliver the Cas9-sgRNA complex in a dose-dependent and transient fashion, we tested whether they could also lead to reduced off-target effects when compared to classical DNA transfection. For this, we developed an approach similar to that described by Fu and colleagues[19] by creating a series of HEK-293T reporter cell lines transduced with different versions of a *GFP* transgene bearing silent point mutations located in the sgRNA target site (Fig. 3b, right panel). These cells were either transfected with plasmids coding for Cas9 and the sgRNA targeting the *GFP* or transduced with Nanoblades programmed with the same sgRNA. 96 h after transfection/transduction, cells were collected and GFP expression was monitored by FACS (Fig. 3b, left panel). As expected, GFP expression from cells bearing the wild-type *GFP* sequence (No Mismatch) was efficiently repressed both after Nanoblades transduction (close to 80% repression) and DNA transfection (close to 60% repression) (Fig. 3b, left panel "No Mismatch"). When two mismatches were introduced in the target site, Nanoblades were no longer able to efficiently repress GFP expression (20% compared to control) while GFP expression from transfected cells was still reduced to levels similar to that of the *GFP* bearing a perfect match with the sgRNA. Interestingly, the presence of three or four mismatches completely abolished *GFP* editing in Nanoblades-treated cells while cells transfected with the Cas9 and sgRNA plasmids still displayed a mild inhibition of GFP expression (Fig. 3b see 3 and 4 Mismatches).

To complement these results, we further tested for genomic off-target effects using the well-characterized sgRNA targeting human *EMX1*. Off-targets for this sgRNA have been extensively studied using T7 endonuclease assays and high-throughput sequencing approaches[10]. We PCR-amplified the *EMX1* locus and one of the previously described *EMX1* genomic off-target loci occurring at the intron of *MFAP1*[10] in cells treated for 72 h with

Nanoblades programmed with the *EMX1* sgRNA or transfected with a DNA construct coding for Cas9 and the *EMX1* sgRNA. We then assessed genome-editing on each sample by high-throughput sequencing (Fig. 3c)[13]. Editing at the on-target site was efficient in Nanoblade-treated cells (75% in average) and to a less extent in cells transfected with the DNA coding for Cas9 and the sgRNA (53% in average) (Fig. 3c, left panel). As expected, small INDELs (insertions and deletions) occurred close to the expected Cas9 cleavage site located 3nt upstream the PAM sequence both in Nanoblades treated and in DNA-transfected cells (Supplementary Figure 4). Surprisingly, in spite of the higher editing efficiency at the on-target site, we could not detect any significant editing at the *MFAP1* off-target site in Nanoblades-treated cells (Fig. 3c, right panel). In contrast, cells transfected with the DNA coding for Cas9 and the sgRNA displayed significant editing (close 6%) at the off-target site (Fig. 3c, right panel) and had INDELs at the expected cut site (Supplementary Figure 4).

Taken together, our results indicate that similarly to other protocols that lead to transient delivery of the Cas9-sgRNA RNP, Nanoblades display low off-target effects.

**Targeted transcriptional activation through Nanoblades.** Having shown efficient genome editing using Nanoblades loaded with the catalytically active Cas9, we tested whether Nanoblades could also deliver Cas9 variant proteins for applications, such as targeted transcriptional activation. To this aim, we fused the Cas9-derived transcriptional activator (SP-dCas9-VPR)[20] to Gag from MLV and expressed the fusion protein in producer cells together with a control sgRNA or different combinations of sgRNAs targeting the promoter region of human Titin (*TTN*) as previously described[20] (Fig. 3d, left panel). Nanoblades loaded with SP-dCas9-VPR were then incubated with MCF-7 cells and induction of TTN measured by quantitative RT-PCR (normalized to GAPDH expression). As observed (Fig. 3d, right panel), when two different sgRNAs were used in combination, TTN transcription was stimulated from 50 to 200 fold compared to the control situation. Interestingly, when combining the four different sgRNAs in a single VLP, we obtained up to 400-fold transcription stimulation of TTN after 4 h of transduction. Our results therefore suggest that in spite of the large molecular size of the SP-dCas9-VPR (predicted at 224 kDa alone and 286 kDa when fused to MLV Gag), neither its encapsidation within VLPs nor its delivery and function within target cells are impaired. The use of Cas9 variants could therefore expand the toolbox of potential applications of Nanoblades in immortalized and primary cells.

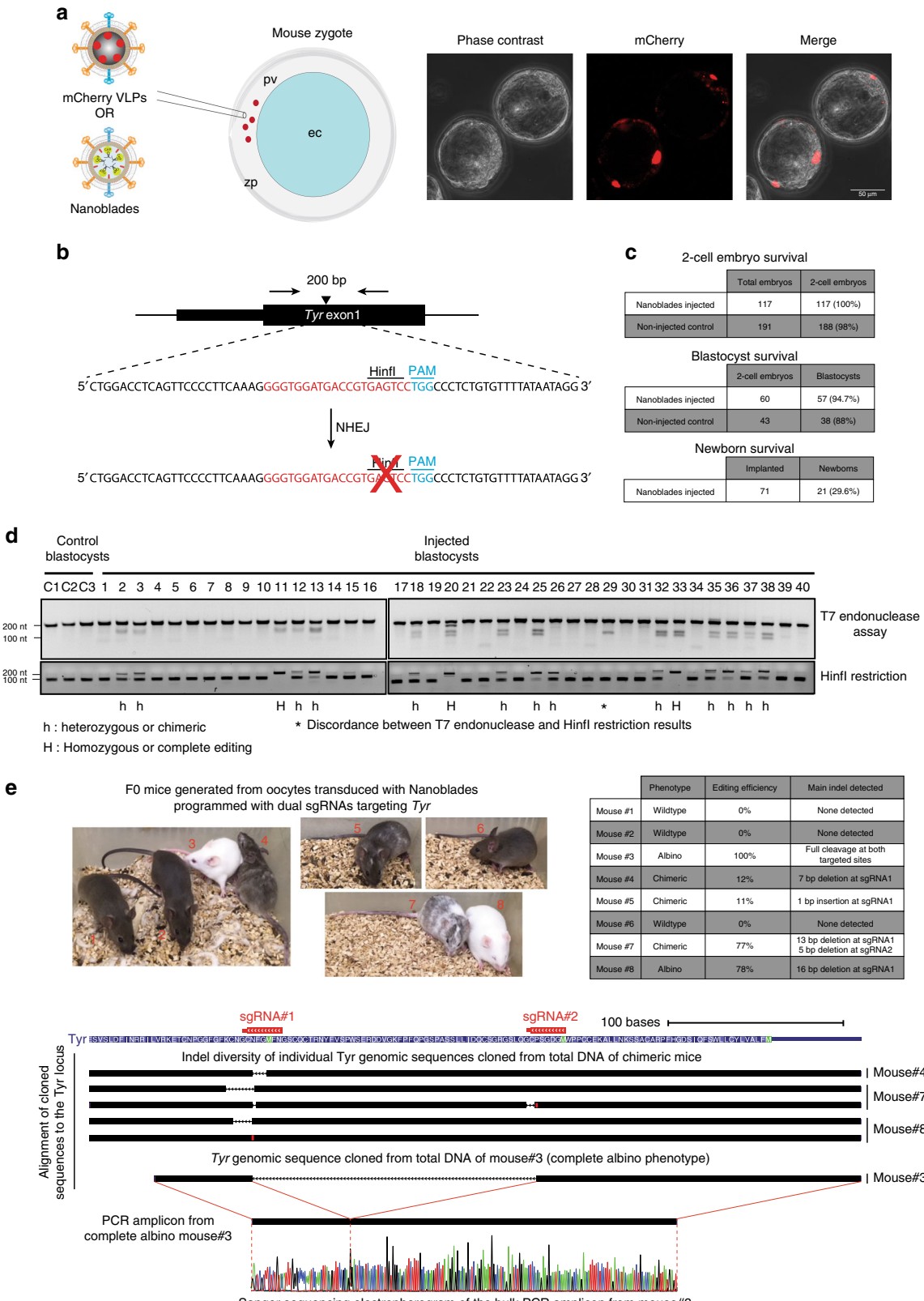

**c**

2-cell embryo survival

|  | Total embryos | 2-cell embryos |
|---|---|---|
| Nanoblades injected | 117 | 117 (100%) |
| Non-injected control | 191 | 188 (98%) |

Blastocyst survival

|  | 2-cell embryos | Blastocysts |
|---|---|---|
| Nanoblades injected | 60 | 57 (94.7%) |
| Non-injected control | 43 | 38 (88%) |

Newborn survival

|  | Implanted | Newborns |
|---|---|---|
| Nanoblades injected | 71 | 21 (29.6%) |

**d**

h : heterozygous or chimeric

H : Homozygous or complete editing

* Discordance between T7 endonuclease and HinfI restriction results

**e** F0 mice generated from oocytes transduced with Nanoblades programmed with dual sgRNAs targeting *Tyr*

|  | Phenotype | Editing efficiency | Main indel detected |
|---|---|---|---|
| Mouse #1 | Wildtype | 0% | None detected |
| Mouse #2 | Wildtype | 0% | None detected |
| Mouse #3 | Albino | 100% | Full cleavage at both targeted sites |
| Mouse #4 | Chimeric | 12% | 7 bp deletion at sgRNA1 |
| Mouse #5 | Chimeric | 11% | 1 bp insertion at sgRNA1 |
| Mouse #6 | Wildtype | 0% | None detected |
| Mouse #7 | Chimeric | 77% | 13 bp deletion at sgRNA1 5 bp deletion at sgRNA2 |
| Mouse #8 | Albino | 78% | 16 bp deletion at sgRNA1 |

Sanger sequencing electropherogram of the bulk PCR amplicon from mouse#3

**Nanoblades-mediated transduction of mouse zygotes**. CRISPR-Cas9 has been extensively used to generate transgenic animals through microinjection of zygotes with DNA coding for Cas9 and the sgRNA or with the synthetic sgRNA and a Cas9 coding mRNA or directly with the preassembled Cas9-sgRNA RNP[21]. However, some of these options usually require injection into the

pronucleus or the cytoplasm of zygotes, which can significantly impact their viability. Moreover, in some species, pronucleus and even cytoplasmic microinjection can be technically challenging.

Because Nanoblades are programmed to fuse with their target cells, we reasoned that they could also transduce murine zygotes without requiring intracellular microinjection. To test this

**Fig. 4** Generation of transgenic mice using Nanoblades. **a** Left panel, scheme describing injection of mCherry VLPs or Nanoblades in the perivitelline space of mouse 1-cell embryos. Right panel, fluorescence microscopy of mouse blastocysts injected with mCherry VLPs at the single-cell stage. **b** Scheme of the design strategy to target the mouse *Tyr* locus (adapted from ref. [22]). Upon editing and NHEJ repair, the HinfI restriction site becomes inactive. **c** Survival rates of injected embryos at two-cell, blastocyst, and newborn stage (the latter obtained from experiments presented in Supplementary figure 5). **d** T7 endonuclease (top panel) and HinfI restrictions (bottom panel) assays on PCR fragments amplified from the *Tyr* locus of Control or Nanoblades-injected embryos. **e** Top left panel, photographs of F0 mice generated from embryos injected with Nanoblades programmed with two sgRNAs targeting the *Tyr* locus. Top-right panel, phenotype, editing efficiency (as measured by TIDE analysis of the Sanger-sequencing electropherograms) and the main INDEL type as detected by Sanger sequencing of individual PCR clones. Bottom-panel, alignment of individual PCR clones obtained from the *Tyr* locus of F0 mice against the mouse mm10 genome indicating the main observed INDELs in chimeric mice (mouse #4, #7, and #8) and total excision of the *Tyr* sequence between the sgRNA1 and sgRNA2 targeting loci for the complete albino mouse (mouse #3). The Sanger sequencing electropherogram from the bulk PCR amplicon obtained from mouse #3 indicates complete editing at both targeted sites

hypothesis, VLPs loaded with the mCherry protein (instead of Cas9) were produced and injected in the perivitelline space of mouse zygotes (Fig. 4a, top panel). Embryos were harvested 80 h after injection (blastocyst stage) and visualized by fluorescence microscopy, showing mCherry protein delivery within embryo cells (Fig. 4a, right panel).

Nanoblades programmed with a sgRNA targeting the first exon of the tyrosinanse (*Tyr*) gene previously described in ref. [22] were produced and injected in the perivitelline space of mouse zygotes. This particular sgRNA was specifically designed to target a HinfI restriction site in the *Tyr* gene that should be disrupted upon dsDNA cleavage and NHEJ repair[22] (Fig. 4b). 80 h after injection, blastocysts were harvested and genomic DNA extracted to monitor genome-editing by PCR amplification followed by T7 endonuclease assay or HinfI restriction. As observed (Fig. 4d), 16 out of 40 blastocysts were positive for genome-editing at the *Tyr* gene both for the T7 endonuclease and the HinfI restriction assays. Interestingly, three blastocysts (#11, #20, and #33) appeared to bear complete *Tyr* editing as we could not detect any residual HinfI restriction products (Fig. 4d). In the remaining 13 blastocysts that were positive for genome editing at the *Tyr* locus, we observed different editing efficiencies thus arguing for variable levels of mosaicism between individuals (Fig. 4d). Interestingly, injection of Nanoblades in the perivitelline was not associated with embryo mortality as we did not obtain any significant difference in survival rates between injected and non-inject embryos (Fig. 4c). To further validate these results, we produced Nanoblades programmed with two sgRNAs targeting the *Tyr* locus (see Fig. 4e bottom scheme) that were injected in the perivitelline space of single-cell embryos, which were then implanted into pseudopregnant females and carried to term. In this case, five out of eight F0 mice obtained carried detectable *Tyr* editing both at the phenotype and genotype level as assayed by PCR amplification of the *Tyr* locus from genomic DNA extracted from the fingers of each animal (Fig. 4e). Interestingly, one of the two fully albino mice carried a complete deletion of the DNA segment between the two sgRNA-targeted loci in all tested cells (as assayed by Sanger sequencing of the bulk PCR product and Sanger sequencing of single clone PCR fragments (Fig. 4e bottom panels)). The remaining F0 mice that displayed a partial *Tyr* disruption phenotype had an editing efficiency ranging from 11% up to 78% of all *Tyr* alleles (Fig. 4e see table). Sanger sequencing of individual PCR clones amplified from these mice indicated that one of the two sgRNAs (sgRNA1) was more efficient in inducing INDELs (Fig. 4e bottom scheme). Moreover, we also detected some degree of mosaicism within each individual mouse (with the exception of mouse #3 which had complete bi-allelic excision of the *Tyr* sequence between the two target loci) with at least two types of INDELs detected in mice 7 and 8 (Fig. 4e, see genomic alignment scheme). This, however, is very similar to the degree of mosaicism found in other approaches[22,23]. Taken together, these results validate the use of

Nanoblades to generate transgenic mice upon perivitelline injection of single-cell embryos.

To further confirm the ability of Nanoblades to mediate genome-editing in mouse embryos and transmission of the edited locus to the offspring, we designed a sgRNA targeting the loxP sequence that could mimic the action of the Cre recombinase by removing a loxP flanked cassette (Supplementary Figure 5, left panel). These Nanoblades were first tested in primary BM cells derived from R26R-EYFP transgenic mice bearing a single-copy of the YFP transgene under control of a "lox-stop-lox" cassette[24] (Supplementary Figure 5, top right panel). Nanoblades were then injected in the perivitelline space of heterozygous R26R-EYFP 1-cell embryos which were then implanted into pseudopregnant females and carried to term. In this case, 1 out of 14 founder animals was YFP positive under ultraviolet (UV) light and displayed efficient excision of the "lox-stop-lox" cassette as confirmed by PCR[25] (Supplementary Figure 5, bottom left panel). Consistent with our previous results, the F1 progeny obtained after mating the loxed F0 mouse with a wild-type mouse contained the "loxed" version of the YFP allele and displayed YFP expression in tails and muscle fibers (Supplementary Figure 5, bottom right panel), indicating efficient transmission of the loxed allele from the F0 founder to its progeny.

Taken together, Nanoblades can represent a viable alternative to classical microinjection experiments for the generation of transgenic animals, in particular for species with fragile embryos or with poorly visible pronuclei.

**In vivo editing of *Hpd* in the liver of tyrosinaemic FRG mice.** Hereditary tyrosinemia type I (HT1) is a metabolic disease caused by disruption of fumarylacetoacetate hydrolase (*Fah*), which is an enzyme required in the tyrosine catabolic pathway. Fah-/- mice recapitulate many phenotypic characteristics of HT1 in humans, such as hypertyrosinemia and liver failure and have to be treated with nitisinone for their survival. Disruption of hydroxyphenylpyruvate dioxigenase (HPD, the enzyme targeted by nitisinone) through hydrodynamic tail vein injection in Fah-/- mice was recently shown to restore their survival in the absence of nitisinone thanks to the selective advantage of Hpd negative hepatocytes[26]. We therefore reasoned that Nanoblades could represent a non-invasive method to inactivate the *Hpd* gene in NRG (NODFah-/-/Rag2-/-/Il2rg-/-) mice[27]. To this aim, we designed a sgRNA directed against the fourth exon of *Hpd*, which should disrupt the reading frame through the INDELs caused by NHEJ (see Methods section for the sequence). Nanoblades directed against *Hpd* or against human *EMX1* (control) were introduced in NRG mice through retro-orbital injection (Fig. 5a). Upon injection, mice were weaned off nitisinone until they reached a 20% loss of their body weight, in which case nitisinone was subsequently administered punctually. Two weeks after injection, all mice injected with Nanoblades targeting *Hpd* displayed detectable editing in the liver (between 7% and 13%

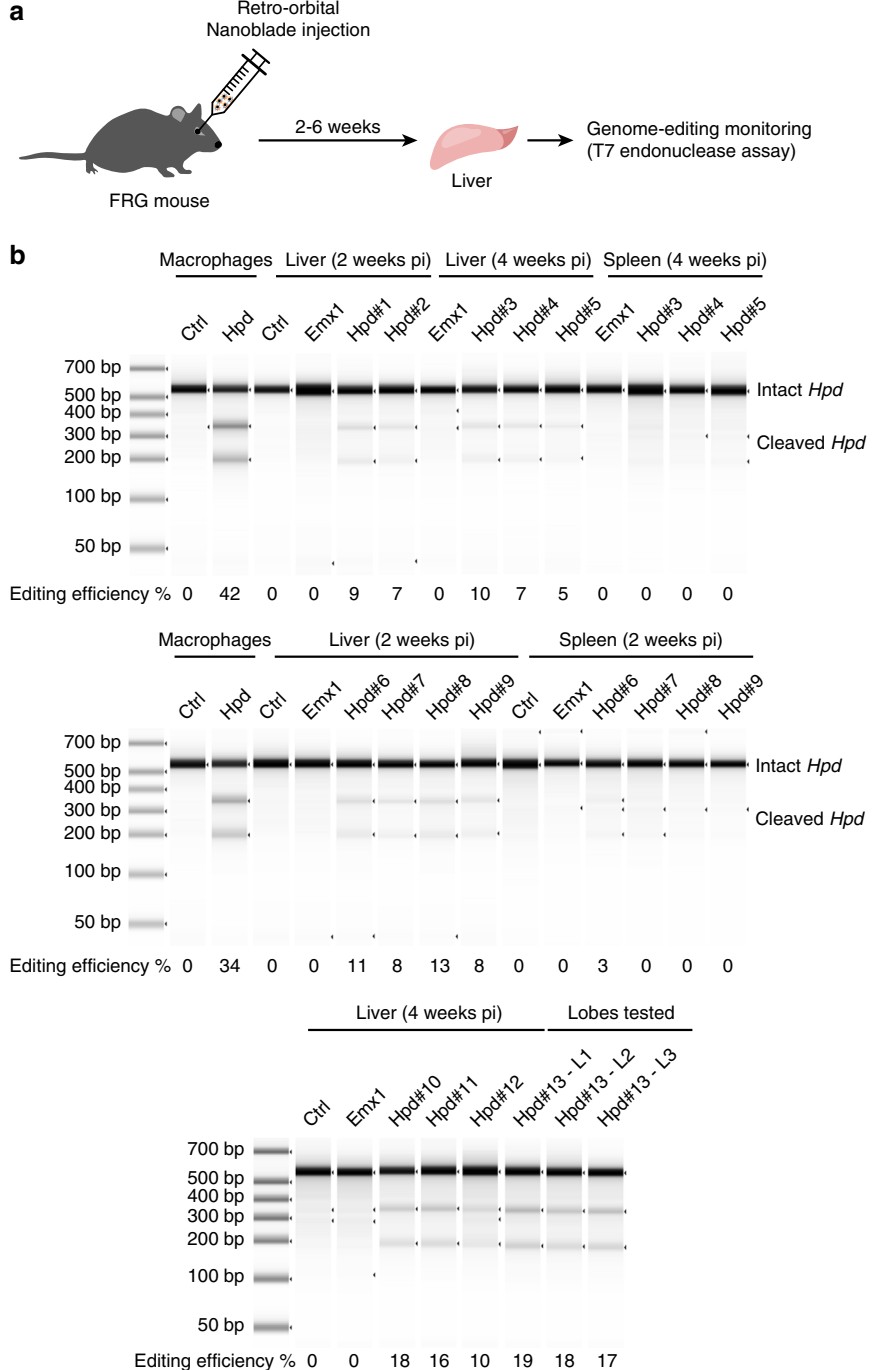

**Fig. 5** Inactivation of *Hpd* in the liver of tyrosinaemic FRG mice. **a** Scheme of the experimental approach to target the liver of FRG mice. **b** T7 endonuclease assay to monitor genome editing at the *Hpd* gene in immortalized mouse macrophages and in the liver or spleen of injected mice. Samples were quantified using a Tapestation chip

efficiency, Fig. 5b). On the contrary, no editing was detected in control (uninjected) mice or in mice injected with Nanoblades targeting human *EMX1* (Fig. 5b). Similar results were obtained 4 weeks post-injection where all mice injected with Nanoblades targeting *Hpd* displayed genome editing in the liver (Fig. 5b). Furthermore, genome-editing occurred in a homogenous fashion across the liver as shown by T7 endonuclease assay from biopsies recovered from three different lobes of a single animal (Fig. 5b, bottom panel). In contrast, editing in other organs, such as spleen was weak or not detectable (Fig. 5b).

Interestingly, we observed a small overall increase in editing levels at 4 weeks post-injection compared to 2 weeks post-

injection suggesting that cells with Hpd editing could have a selective advantage over non-edited cells (Fig. 5b compare middle and bottom panel). Because we did not monitor genome editing earlier than 2 weeks post injection, we cannot rule out that a similar selective advantage of edited cells might have occurred during this incubation time. Nevertheless, based on the weak increase of the editing efficiency observed between 2 and 4 weeks after injection, we do not expect this selective advantage to significantly improve the observed editing efficiency during the first 2 weeks after injection. Importantly, Nanoblades injection was not associated with any signs of morbidity.

## Discussion

Genome editing should ideally be achieved in a fast and precise fashion to limit toxicity and possible off-target effects due to a sustained expression of effectors. In this regard, extensive efforts have been recently described to vehicle Cas9-sgRNA RNPs in cultured cells and in vivo by non-coding material including Nanocarriers[28], optimized transfection reagents[18], or lentivirus-derived particles[29].

This work describes and characterizes VLPs to efficiently vectorize the CRISPR-Cas9 system into primary cells, embryos, and animals. These non-coding agents—we called herein Nanoblades—incorporate the Cas9 endonuclease into their internal structure. The molecular basis of this technology is the fusion of Cas9 from *Streptococcus pyogenes* to Gag from MLV. Expressed with other components of viral assembly and construct encoding gRNA(s), this molecule can bind sgRNAs into producer cells, forms RNP complexes and cohabit with Gag and Gag-Pol within particles. We indeed show that robust packaging of sgRNAs into Nanoblades depends on their interaction with Gag::Cas9 (Supplementary Figure 1d).

When compared to other methods of delivery such as lipofection or electroporation, Nanoblades were more efficient and rapid in inducing dsDNA breaks both in immortalized U2OS cells, primary fibroblasts (Fig. 1b, Supplementary Figure 1e). Nanoblades are also functional in primary cells that are known to be difficult to transfect and transduce using classical delivery methods, such as human iPS cells, human CD34+ and primary mouse bone-marrow cells (Fig. 2) reaching efficiencies comparable or even superior to other recent methods[30,31], such as Cas9-sgRNA ribonucleoprotein electroporation (Fig. 2c), together with low off-target effects (Fig. 3b and c). Furthermore, Nanoblades achieve genome editing in a dose-dependent manner (Fig. 1c). Beyond delivery of Cas9-sgRNA complexes, we also show that Nanoblades can be complexed with DNA repair templates to mediate homologous recombination-based knock-in cultured cells in the absence of any transfection reagent. Our results also validate the use of Nanoblades in vivo for generating transgenic mice upon embryo injection in the perivitelline space (Fig. 4 and Supplementary Figure 5) or in the liver of injected animals (Fig. 5). Although, other recent methods for in vivo genome editing of zygotes and animals have reached higher editing rates[22,23,32–34], Nanoblades represent a viable, inexpensive, and accessible alternative that can still benefit from further improvements.

Similarly to other cell-derived particles (including most viral vectors), Nanoblades incorporate RNAs and proteins from producer cells that could be responsible for the transmission of undesired effects. Mass spectrometry analysis of the content of Nanoblades revealed that plasma membrane terms were particularly enriched, which is consistent with the vesicular nature of Nanoblades (Supplementary Figure 2a and Supplementary Data 1). As previously described for retroviral-VLPs[35], characterization of the RNA content revealed that Nanoblades contain thousands of individual cellular mRNA species, most of these being encapsidated stochastically, in proportion to their abundance in the producer cell. We found that transcripts over-expressed for production purposes (GAG, VSV-G, etc.) represent <0.4% of Nanoblades RNAs (Supplementary Figure 2b) supporting the notion that their delivery to recipient cells is marginal. Confirming this observation, transfer of cellular proteins loaded in Nanoblades from producer cells to recipient cells appears to be minimal and restricted to a short time window between 8 and 24 h after transduction (Supplementary Figure 2c and d). While we cannot exclude the fact that VLPs may be responsible for some cellular responses, depending on the nature of recipient cells, efficient doses of Nanoblades were globally harmless for most

primary cells we tested and in injected animals. In our effort to exploit the retroviral nature of Nanoblades, we explored diverse pseudotyping options (Supplementary Figure 6) and finally focused on the use of an original mixture of two envelopes (VSV-G plus BRL), a recipe that we have optimized (Supplementary Figure 6) and which systematically displayed the best cleavage results in most recipient cells. Depending on the cellular target, it may be possible to pseudotype Nanoblades with envelopes from Measles virus[36], influenza virus[37], or other targeting systems[38,39] to restrict or improve Cas9 delivery to certain cell types (Supplementary Figure 6a).

Next generation Nanoblades may also benefit from the continual evolutions of Cas9-derivatives that can support fusion with Gag from MLV (Fig. 3) and could be adapted to other gene-editing targetable nucleases like Cpf1 nucleases[40] or even the latest generation of programmable base editors[41]. We also noted that Nanoblades can be engineered to accommodate other proteins/RNAs in addition to Cas9-RNPs and serve as multifunctional agents. Nanoblades capable of delivering both Cas9-RNPs and a reverse-transcribed template that can serve for reparation by homologous-recombination could therefore be envisioned. Furthermore, multiple sgRNAs can be incorporated within Nanoblades thus allowing gene excisions or multiple genes to be targeted. Multiplexing of sgRNAs may also allow the introduction of an additional sgRNA targeting a specific gene that will allow selection of cells efficiently edited by Nanoblade-mediated CRISPR[42].

This versatility allows any laboratory equipped with BSL2 facilities to generate its own batches of particles. Beyond cell lines, our VLP-based technique provides a powerful tool to mediate gene editing in hiPSCs and primary cells including macrophages, human hematopoietic progenitors and primary hepatocytes. We have shown that Nanoblades injection into the perivitelline space of mouse-zygotes was particularly harmless for the recipient cells, since none of the injected zygotes were affected in their development after treatment. Generation of transgenic animals upon perivitelline space injection of VLPs could be adapted to other species, including larger animals for which the number of zygotes is limited. Finally, we achieved significant gene-editing in the liver of injected adult mice with no consequences on their viability. Nanoblades, could therefore represent an interesting route for the delivery of Cas9 in vivo to inactivate gene expression but also used in combination with other viral delivery tools carrying a donor DNA template (such as Adeno-associated virus (AAV)) to perform in vivo HDR experiments as recently shown[32].

Considering the examples provided in our work, we believe that the Nanoblade technology will facilitate gene editing in academic laboratories working with primary cells and could represent a viable alternative for therapeutic purposes and the rapid generation of primary cell-types harboring genetic diseases, humanized-liver mouse models and transgenic animal models.

## Methods

**Plasmids.** SP-dCas9-VPR was a gift from George Church (Addgene plasmid #63798). Lenti CRISPR was a gift from F. Zhang (Addgene plasmid #49535). The GagMLV-CAS9 fusion was constructed by sequential insertions of PCR-amplified fragments in an eukaryotic expression plasmid harboring the human cytomegalovirus early promoter (CMV), the rabbit Beta-globin intron and polyadenylation signals. The MA-CA-NC sequence from Friend MLV (Accession Number: M93134) was fused to the MA/p12 protease-cleavage site (9 aa) and the Flag-nls-spCas9 amplified from pLenti CRISPR.

**Cell culture.** Gesicle Producer 293T (Clontech 632617), U2OS cells, and primary human fibroblasts (Coriell Institute, GM00312) were grown in DMEM supplemented with 10% fetal calf serum (FCS).

hiPSCs were obtained and cultured as described in ref. [43].

Bone marrow-derived macrophages (BMDMs) were differentiated from BM cells obtained from wild-type C57BL/6 mice. Cells were grown in DMEM

supplemented with 10% FCS and 20% L929 supernatant containing MCSF as described in ref. [44]. Macrophages were stimulated for the indicated times with LPS (Invivogen) at a final concentration of 100 ng/ml.

**CD34+-cell sample collection, isolation, and transduction.** Cord blood (CB) samples were collected in sterile tubes containing the anti-coagulant, citrate-dextrose (ACD, Sigma, France) after informed consent and approval was obtained by the institutional review board (Centre international d'infectiologie (CIRI), Lyon, France) according to the Helsinki declaration. Low-density cells were separated over, Ficoll-Hypaque. CD34+ isolation was performed by means of positive selection using magnetic cell separation (Miltenyi MACs) columns according to the manufacturer's instructions (Miltenyi Biotec, Bergisch Gladbach, Germany). Purity of the selected CD34+ fraction was assessed by FACS analysis with a phycoery-thrin (PE)-conjugated anti-CD34 antibody (Miltenyi Biotec, Bergisch Gladbach, Germany) and exceeded 95% for all experiments. Human CD34+ cells were incubated for 18–24 h in 24-well plates in serum-free medium (CellGro, CellGenix, Germany) supplemented with human recombinant: SCF (100 ng/ml), TPO (20 ng/ml), Flt3-L (100 ng/ml) (Myltenyi, France). $5 \times 10^4$ prestimulated CD34+ cells were then incubated with nanoblades in 48-well plates in serum-free medium.

**sgRNA design and sequences (+PAM).** sgRNAs targeting *MYD88, DDX3, GFP, Hpd, Fto, Tyr*, and the *LoxP* sequence were designed using CRISPRseek[45].

Human *AAVS1*: 5′ ACCCCACAGTGGGGGCCACTAggg 3′
Human *DDX3*: 5′ AGGGATGAGTCATGTGGCAGtgg 3′
Human *EMX1*: 5′ GAGTCCGAGCAGAAGAAGAAggg 3′
Human *MYD88* #1: 5′ GAGACCTCAAGGGGTAGAGGTggg 3′
Human *MYD88* #2: 5′ GCAGCCATGGCGGGGCGGTCCtgg 3′
Human rDNA: 5′ CCTTCTCTAGCGATCTGAGagg 3′
Human *TTN* -169: 5′ CCTTGGTGAAGTCTCCTTTGagg 3′
Human *TTN* -252: 5′ ATGTTAAAATCCGAAAATGCagg 3′
Human *TTN* -326: 5′ GGGCACAGTCCTCAGGTTTGggg 3′
Human *TTN* -480: 5′ ATGAGCTCTCTTCAACGTTAagg 3′
Mouse *Fto*: 5′ CATGAAGCGCGTCCAGACCGcgg 3′
Mouse *Hpd*: 5′ GAGTTTCTATAGGTGGTGCTGGGTGggg 3′
Mouse *Tyr*: 5′ GGGTGGATGACCGTGAGTCCtgg 3′ obtained from Chen et al. [22]
Mouse Tyr: 5′ AACTTCATGGGTTTCAACTGcgg 3′ obtained from Yoon et al. [23]
Mouse *Tyr*: 5′ ATGGGTGATGGGAGTCCCTGcgg 3′ this study
*LoxP*: 5′ CATTATACGAAGTTATATTAagg 3′
*GFP*: 5′ CGAGGAGCTGTTCACCGGGGtgg 3′

**Production of Nanoblades.** Nanoblades were produced from transfected gesicles producer 293T cells plated at $5 \times 10^6$ cells/10 cm plate 24 h before transfection with the JetPrime reagent (Polyplus). Plasmids encoding the GagMLV-CAS9 fusion (1.7 µg), Gag-POLMLV (2.8 µg), gRNA expressing plasmid(s) (4.4 µg), VSV-G (0.4 µg), the Baboon Endogenous retrovirus Rless glycoprotein (BaEVRless)[15] (0.7 µg) were cotransfected and supernatants were collected from producer cells after 40 h. For production of serum-free particles, medium was replaced 24 h after transfection by 10 ml of Optimem (Gibco) supplemented with penicillin–streptomycin. Nanoblade-containing medium was clarified by a short centrifugation ($500 \times g$ 5 min) and filtered through a 0.8 µm pore-size filter before ultracentrifugation (1h30 at $96,000 \times g$). Pellet was resuspended by gentle agitation in 100 µl of cold 1X PBS. Nanoblades were classically concentrated 100-fold. X-Nanoblades referred as Nanoblades loaded with gRNA(s) targeting the x-gene.

To dose Cas9 packaged into particles, Nanoblades or recombinant Cas9 (New England Biolabs) were diluted in 1X PBS and serial dilutions were spotted onto a Nitrocellulose membrane. After incubation with a blocking buffer (nonfat Milk 5% w/v in TBST), membrane was stained with a Cas9 antibody (7A9-3A3 clone, Cell signaling) and revealed by a secondary anti-mouse antibody coupled to horseradish peroxidase. Cas9 spots were quantified by Chemidoc touch imaging system (Biorad).

**Transduction procedure.** Transductions with Nanoblades were performed in a minimal volume to optimize cell/particles interactions for at least 2 h before supplementing with fresh medium. When specified, polybrene was used at a final concentration of 4 µg/ml in the transduction medium. After dosing Cas9 amount in each Nanoblades preparation, we typically used 10 pmol of encapsidated Cas9 for $1 \times 10^5$ adherent cells.

**sgRNA in vitro transcriptions.** sgRNAs were in vitro transcribed using the EnGen sgRNA Synthesis kit, *S. pyogenes* (New England Biolabs; E3322S) following the manufacturer's protocol with the following oligonucleotides:
Human *EMX1*: 5′ TTCTAATACGACTCACTATAgagtccgag cagaagaagaaGTTTTAGAGCTAGA 3′
Mouse *Fto*: 5′ TTCTAATACGACTCACTATAgcatgaagcgcgtc cagaccgGTTTTAGAGCTAGA 3′

After transcription, sgRNAs were purified by acidic phenol/chloroform extraction and precipitated using 2.5 volumes of 100% ethanol. sgRNA integrity was then assessed by denaturing urea polyacrylamide gel electrophoresis.

**Cas9-sgRNA RNP electroporation procedure.** Cas9-sgRNA RNP electroporation was performed as described in the manufacturer's protocol. Briefly, 12 pmol of recombinant Cas9 (EnGen Cas9 NLS, S. pyogenes; New England Biolabs; M0646T) were incubated with 12 pmol of in vitro transcribed sgRNAs in the presence of Resuspension Buffer R (Neon Transfection System; ThermoFisher Scientific; MPK1025) for 20 min at room temperature. After this, $1 \times 10^5$ cells resuspended in 5 µl of resuspension buffer R (for HEK293T cells) or resuspension buffer T (for primary mouse BM cells) are added to the Cas9-sgRNA mix and the whole mixture electroporated with the following settings:
-1700 V, 20 ms, 1 pulse (HEK293T cells)
-1350 V, 10 ms, 4 pulses (mouse BM cells)
-1680 V, 20 ms, 1 pulse (mouse BM cells)
Upon electroporation, cells were incubated in their corresponding medium (DMEM complemented with 10% FCS for HEK293T cells and DMEM complemented with 10% FCS and 20% L929 supernatant containing MCSF for 24 h before extracting their genomic DNA to assess genome editing.

**Combination of Nanoblades with ssDNA and dsDNA.** Nanoblades programmed to target the AUG codon of *DDX3* were resuspended in PBS 2% FBS and combined with ssDNA donor repair template (see the sequence of "Flag-DDX3 primer" below) at a final concentration of 0.3, 1.3 or 6.7 µM in 30 µl of PBS supplemented with polybrene (Sigma) at 4 µg/ml. Complexes were let 15 min on ice before addition to $7 \times 10^4$ HEK293T cells plated 6 h before in 400 µl of complete medium supplemented with polybrene (4 µg/ml). 24 h latter, transduction medium was supplemented with 1 ml of fresh medium (10% FCS) and cells were passed the day after into six-well plates for amplification. Cells were amplified in 10 cm dishes and passed six times during 3 weeks before extraction of proteins and genomic DNAs.

Sequence of the Flag-DDX3 primer (HPLC-purified):
5′-ACTCGCTTAGCAGCGGAAGACTCCGagTTCTCGGTA CTCTTCAGGGATGGA
CTACAAGGACGACGATGACAAGagTCATGTGGCAGTG GAAAATGCGCTCGGGGCTGGACCAGCAGGTGA-3′

DDX3 amplification was performed using the following primers: DDX3-Forward 5′-CTTCGCGGTGGAACAAACAC-3′ and DDX3-Reverse1 5′-CGCCATTAGCCAGGTTAGGT-3′ for the "Insertion PCR assay" and Flag-Forward 5′-GACTACAAGG ACGACGATGACAAG-3′ and DDX3-Reverse2 5′-CGCCATTA GCCAGGTTAGGT-3′ for the "Orientation PCR assay". PCR conditions were performed as follows: 94 °C 5 min, followed by three cycles (94 °C 30 s, 64 °C 30 s, 72 °C 30 s), followed by 25 cycles (94 °C 30 s, 57 °C 30 s, 72 °C 30 s), followed by 5 min at 72 °C.

dsDNA (AAVS1): 10 µl of concentrated Nanoblades were complexed with 650 ng of dsDNA in a total volume of 30 µl of PBS with polybrene at a final concentration of 4 µg/ml. After 15 min of incubation on ice, complexes were used to transduce $1 \times 10^5$ HEK293T cells in a 24-well plate containing medium supplemented with polybrene (4 µg/ml). Two days latter cells were reseeded in a 10 cm dish before puromycin selection (0.5 µg/ml). Single-cell-derived clones were next isolated and cultivated in a 12-well plates before PCR analysis performed on genomic DNAs (500 ng).

Primers used to assess the presence of the puromycin cassette are:
Puromycin-forward 1: 5′-GGCAGGTCCTGCTTTCTCTGAC-3′
Puromycin-reverse 1: 5′-GATCCAGATCTGGTGTGGCGCG TGGCGGGGTAG-3′
Followed by a nested-PCR using the following primers:
Puromycin-forward 2: 5′-GATATACGCGTCCCAGGGCCGG TTAATGTGGCTC-3′
Puromycin-reverse 1: 5′-GATCCAGATCTGGTGTGGCGCG TGGCGGGGTAG-3′
Primers used to assess correct integration of the cassette at the AAVS1 locus are:
AAVS1-forward: 5′-CGGAACTCTGCCCCTCTAACGCTG-3′
Puromycin reverse 2: 5′-GATCCAGATCTGGTGTGGCGCG TGGCGGGGTAG-3′
Followed by a nested-PCR using the following primers:
AAVS1-forward: 5′-GGCAGGTCCTGCTTTCTCTGAC-3′
Puromycin reverse 3: 5′-CACCGTGGGCTTGTACTCGGT CAT-3′

**Flag-immunoprecipitation and western-blotting.** For Flag-immunoprecipitation, $5 \times 10^6$ cells were lysed in 500 µl of lysis buffer (NaCl 300 mM, MgCl₂ 6 mM, Tris–HCl 15 mM, 0.5% NP40). 250 µl of the cell lysate (1 mg of total proteins) was incubated with 40 µl of M2-antiFlag magnetic beads (Sigma M8823) equilibrated in TBS. After incubation for 2 h at 4 °C, beads were washed four times in lysis buffer and proteins eluted in 60 µl of TBS supplemented with Flag-peptide (120 µg/ml final) for 2 h at 4 °C. The supernatant (without beads) was then collected and used for western-blot analyses.

Western-blotting against Flag-DDX3 and endogenous DDX3 was performed using the following antibodies: anti-DDX3 (rabbit, Sigma 19B4, 1/1000 dilution), Flag-M2 Antibody (mouse, Sigma F3165, 1/2000 dilution), and actin antibody (mouse, Sigma A1978, 1/10,000 dilution). The uncropped images for

Supplementary Figs. 1a, 2d, 3d and 2b–d, 3a, 4d are provided in Supplementary Fig. 7.

**T7 endonuclease assay.** Genomic DNA was extracted from VLP-treated cells using the Nucleospin gDNA extraction kit (Macherey-Nagel). 150 ng of genomic DNA was then used for PCR amplification. PCR products were diluted by a factor 2 and complemented with Buffer 2 (New England Biolabs) to a final concentration of 1×. Diluted PCR amplicons were then heat denatured at 95 °C and cooled down to 20 °C with a 0.1 °C/s ramp. Heteroduplexes were incubated for 30 min at 37 °C in presence of 10 units of T7 Endonuclease I (NEB). Samples were finally run on a 2.5% agarose gel or on a BioAnalyzer chip (Agilent) to assess editing efficiency.

**Reverse-transcription and quantitative PCR.** Total RNAs were extracted using TriPure Isolation Reagent (Roche, 11667165001) following the manufacturer's instructions. 1.5 μg of total RNA was treated with DNase and reverse-transcribed using Maxima First Strand cDNA Synthesis Kit for RT-qPCR (Thermo Scientific, K1672) following the manufacturer's instructions. qPCR experiments were performed on a LightCycler 480 (ROCHE) in technical triplicates in 10 μl reaction volume as follows: 5 μl of 2X SYBR qPCR Premix Ex Taq (Tli RNaseH Plus) (TAKARA, TAKRR420W); forward and reverse primers (0.5 μM each final); 7.5 ng of cDNA.

**Immunofluorescence and imaging.** Cells were fixed in 1X PBS supplemented with 4% of paraformaldehyde (PFA) for 20 min, washed three times with 1X PBS and permeabilized with 0.5% Triton X-100 for 4.5 min. Cells were incubated with primary antibodies overnight at 4 °C. Primary antibodies used are: rabbit yH2AX (1:1000; Abcam 81299) and mouse RNA pol I RPA194 (1:500; Santacruz sc48385). Cells were washed three times in 1X PBS, followed by incubation of the secondary antibodies conjugated to Alexa 488 or 594 used at a 1:1000 dilution (Life Technologies) for 1 h at room temperature. After three 1X PBS washes, nucleus were stained with Hoechst 33342 at 1 μg/ml for 5 min. The coverslips were mounted in Citifluor medium (AF1, Citifluor, London, UK). Cells were observed under a Leica DM6000. At least 100 cells were counted in each indicated experiment. Averages and standard deviation values were obtained from three independent biological replicates.

**Flow cytometry analysis of CD81 expression.** $1 \times 10^6$ HepG2 or HEK293T cells were detached from the cell culture plate using Accutase (Stemcell technologies #07920) and washed twice in PBS + 2%BSA. Cells were then incubated in 100 μl of PBS + 2%BSA + Anti-CD81 (BD Biosciences #555675, clone JS-81, 1/200 dilution) for 30 min at 4 °C. Cells were then washed three times in PBS + 2% BSA and incubated in 100 μl of PBS + 2 %BSA + anti-mouse FITC (Biolegend # 406001, 1/2000 dilution) for 30 min at 4 °C in the dark. Cells were then washed three times in PBS + 2%BSA and fixed with 4% of paraformaldehyde (PFA) for 15 min and washed in PBS + 2%BSA before flow cytometry analysis on a BD FACSCanto II.

**Northern-blot of sgRNAs.** 2 μg of total RNA extracted from Nanoblades or Nanoblade-producing cells were run on a 10% acrylamide, 8 M Urea, 0.5X TBE gel for 1 h at 35 W. RNAs were then transferred onto a Nitrocellulose membrane (Hybond Amersham) by semi-dry transfer for 1 h at 300 mA in 0.5X TBE. The membrane was UV-irradiated for 1 min using a stratalinker 1800 and then baked at 80 °C for 30 min. The membrane was then incubated in 50 ml of Church buffer (125 mM $Na_2HPO_4$, 0.085% phosphoric acid, 1 mM EDTA, 7% SDS, 1% BSA) and washed twice in 10 ml of Church buffer. The 5′ P32-labeled ($1 \times 10^7$ cpm total) and heat-denatured ssDNA probe directed against the constant sequence of the guideRNA (sequence of the sgRNA antisense probe: 5′GCACCGACTCGGTGCCA CTTTTTCAAGTTGATAACGGACTAGCCTTATTTTAACTTGCTATTTCTA GCTCTA3′) was diluted in 10 ml of Church buffer and incubated with the membrane overnight at 37 °C. The membrane was washed four times in 50 ml of wash buffer (1X SSC + 0.1% SDS) before proceeding to phosphorimaging.

**Transmission electron microscopy (TEM) and mass spectrometry (MS).** Nanoblades programmed to target the YFP were prepared and processed for TEM and MS as previously described[46]. Briefly, Nanoblades were produced from transfected Gesicles Producer 293T cells plated at $5 \times 10^6$ cells/10 cm plate 24 h before transfection with the JetPrime reagent (Polyplus) and supernatants were collected from producer cells after 40 h, passed through a 0.45 μm filter and concentrated 100-fold by overnight centrifugation at $3800 \times g$. This preparation was next laid overlaid on a continuous optiprep gradient and ultracentrifuged to obtain density fractions. Fractions containing Nanoblades were next pooled and centrifuged overnight at $3800 \times g$ before PBS resuspension to obtain a 6000×-concentrated sample.

For electron microscopy, after a flash-fixation in glutaraldehyde, staining was amplified using the R-Gent Kit (Biovalley, Marne-la-Vallee, France) before the negative coloration (phosphotungstic acid 2%). Specimen were observed under a JEM-1400 microscope (Jeol, Tokyo, Japan) coupled to the Orius-600 camera (Gatan, Pleasanton, CA).

**High-troughput sequencing of RNAs extracted from Nanoblades.** Total RNA was extracted from purified Nanoblades programmed to target the YFP using Trizol. RNAs were then fragmented to 100nt and used as input for the preparation of cDNA libraries following the protocol described in ref. [47]. Briefly, RNA fragments with a 3′-OH were ligated to a preadenylated DNA adaptor. Following this, ligated RNAs were reverse transcribed with Superscript III (Invitrogen) with a barcoded reverse-transcription primer that anneals to the preadenylated adaptor. After reverse transcription, cDNAs were resolved in a denaturing gel (10% acrylamide and 8 M urea) for 1 h and 45 min at 35 W. Gel-purified cDNAs were then circularized with CircLigase I (Epicentre) and PCR-amplified with Illumina's paired-end primers 1.0 and 2.0.

Analysis of high-troughput sequencing data was performed as previously described[48]. Briefly, reads were split with respect to their 5′-barcode sequence. After this, 5′-barcode and 3′-adaptor sequences were removed from reads. Reads were mapped to a custom set of sequences including 18S, 28S, 45S, 5S, and 5.8S rRNA, tRNAs, the sgRNA directed against the GFP sequence and all transcripts coding for Nanoblades components (Envelopes, Gag and Pol, Cas9) using Bowtie[49]. Reads that failed to map to this custom set of sequences were next aligned to University of California, Santa Cruz (UCSC) human hg18 assembly using TopHat[50]. Read counts on all transcripts of interest were obtained using the HTSeq count package[51].

**High-throughput sequencing of Emx1 On-target and Off-target loci.** Genomic DNA was extracted from Nanoblades-treated cells using the Nucleospin gDNA extraction kit (Macherey-Nagel). 150 ng of genomic DNA was then used for PCR amplification using primers specific for the *EMX1* On-target locus (EMX1-Forward 5′-ACACTCTTTCCCTACACGACGCTCTTCCGATCTGGTTCCAGAACCGG AGGACAAAGTAC-3′ and EMX1-Reverse 5′-GTGACTGGAGTCCTCTCTAT GGGCAGTCGGTGAAGCCCATTGCTTGTCCCTCTGTCAATG-3′) and the previously described Off-target locus in the intron of *MFAP1* (MFAP1-Forward 5′- ACACTCTTTCCCTACACGACGCTCTTCCGATCTCCATCACGGCCTTTG CAAATAGAGCCC-3′ and MFAP1-Reverse 5′-GTGACTGGAGTCCTCTCTA TGGGCAGTCGGTGACAGAGGGAACTACAAGAATGCCTGAGC-3′) bearing adapters sequencing for Illumina's Miseq platform. Obtained PCR products were purified and PCR amplified with a second set of primers bearing specific barcodes for multiplex sequencing. Final PCR products were sequenced on the Miseq platform using a custom sequencing primer (Miseq-Custom 1: 5′ ATCACCGACTGCCCATAGAGAGGACTCCAGTCAC 3′) and a custom index sequencing primer (Miseq-Custom 2: 5′ GTGACTGGAGTCCTCTCTATGGGC AGTCGGTGAT 3′).

**Animal experimentation.** All animal experiments were approved by a local ethics committee of the Université de Lyon (CECCAPP, registered as CEEA015 by the French ministry of research) and subsequently authorized by the French ministry of research (APAFIS#8154-2016l12814462837 v2 for the generation of transgenic animals and C 69 123 0303 for the usage of Nanoblades in vivo). All procedures were in accordance with the European Community Council Directives of September 22, 2010 (2010/63/EU) regarding the protection of animals used for scientific purposes.

**Mouse oocyte injection.** Four or five weeks old FVB/NRj female mice (Janvier Labs, France) were superovulated by intraperitoneal (i.p.) administration of 5 IU of pregnant mare serum gonadotropin (PMSG, Alcyon, France), followed by an additional i.p. injection of 5 IU human chorion gonadotropin 48 h later (hCG, Alcyon, France). Superovulated females were mated with B6D2F1 adult males (1 male/2 females) and euthanatized at 0.5 day post coitum (usually between 10 and 11 a.m.). Oviduct were dissected, and the ampulla nicked to release zygotes associated with surrounding cumulus cells into a 200 μl droplet of hyaluronidase (Sigma) in M2 solution (300 μg/ml, Sigma) under a stereomicroscope (Olympus SZX9). Zygotes were incubated for 1 min at room temperature and passed with a mouth pipette through three washes of M2 medium to remove cumulus cells. Zygotes were kept in M16 medium (Sigma) in a water jacketed $CO_2$ incubator (5% $CO_2$, 37 °C) until microinjection with Nanoblades. Micro-injection were carried-out under a stereomicroscope (Olympus SZX9) using a FemtoJet 4i (Eppendorf) microinjecter. Briefly, 1 pl of Nanoblades were injected in the perivitelline space of oocytes. Zygotes were then transferred into M16 medium and kept overnight in incubator. The embryos that reached the two-cell stage were transferred into the oviduct of B6CBAF1 (Charles River, France) pseudopregnant females (15–20 embryos per female).

**Retro-orbital injection of Nanoblades.** All experiments were performed in accordance with the European Union guidelines for approval of the protocols by the local ethics committee (Authorization Agreement C2EA 15, "Comité Rhône-Alpes d'Ethique pour l'Expérimentation Animale", Lyon, France). The highly Immunosuppressed NOD FRG mice (Fah-/-/Rag2-/-/Il2rg-/-) (Yecuris cooration), deficient for T-cell, B-cell, and NK-cell are maintained in pathogen-free facility. Retro-orbital injection (SRO) were performed under isoflurane anesthésia.

Genomic DNA from each mouse (treated either by control or *Hpd* targeting Nanoblades) was extracted from three distinct liver lobes and pooled together.

Following this, a two-step PCR was performed on 300 ng of gDNA template, the first PCR using primers Hpd-Forward 1: 5′-CTTAGGAGGTTAGCCAAAGATG GGGAG-3′ and Hpd-Reverse 1: 5′-TCTAGTCTCTATCCAGGGCTCCAGCC-3′ to amplify the *Hpd* gene (94 °C 5 min, 3 cycles 94 °C, 64 °C, 72 °C, and 20 cycles 94 °C, 58 °C, 72 °C, 5 min 72 °C). The second nested-PCR used primers Hpd-Forward 2: 5′-GAACTGGGATTGGCTAGTGCG-3′ and Hpd_Reverse 2: 5′-CACCCAG CACCACCTATAGAAACTC-3′ (94 °C 5 min, 3 cycles 94 °C, 64 °C, 72 °C and 30 cycles 94 °C, 57 °C, 72 °C, 5 min 72 °C). Amplicons were next analyzed by T7-endonuclease assay as described.

**Raw data files.** Uncropped scans of ethidium bromide gels and western-blotting figures are displayed in Supplementary Figure 7.

## Data availability

Gene Expression Omnibus: GSE107035. The following plasmids will be available from Addgene: Gag::Cas9 fusion (BIC-Gag-CAS9, Plasmid ID: 119942), the Gag::Cas9-VPR fusion (BICstim-Gag-dCAS9-VPR, Plasmid ID: 120922) and the Gag::Cre fusion (GAG-CRErec, Plasmid ID: 119971).

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

## Acknowledgements

Sequencing was performed by the IGBMC Microarray and Sequencing platform, a member of the 'France Génomique' consortium (ANR-10-INBS-0009). We acknowledge the contribution of SFR Biosciences (UMS3444/CNRS, US8/Inserm, ENS de Lyon, UCBL) facilities: Platim and PBES (Celphedia, AniRA). We also thank J.F. Henry, N. Aguilera, and J.L. Thoumas from the animal facility (PBES, Plateau de Biologie Expérimentale de la Souris, ENS de Lyon), as well as A. Ollivier for their technical help in handling mice. We thank Claire Lionnet from Platim for technical assistance in taking confocal fluorescence images. We thank Elisabeth Errazuriz-Cerda and the CeCIL-facility (Lyon, France) for the preparation and the observation of samples by TEM and Yohann Couté and the edyp-service (Grenoble, France) for the proteomic analysis. We thank Gérard Benoît for his help in preparing final figures. This work was funded by Labex Ecofect (ANR-11-LABX-0048) of the Université de Lyon, within the program Investissements d'Avenir (ANR-11-IDEX-0007) operated by the French National Research Agency (ANR), Fondation FINOVI and Agence Nationale des Recherches sur le SIDA et les Hépatites Virales (ANRS—ECTZ3306) to E.P.R. Open access fees were funded by the European Research Council (ERC-StG-LS6-805500 to E.P.R.) under the European Union's Horizon 2020 research and innovation programs.

## Author contributions

P.E.M. and E.P.R. conceived the study and designed most experiments. P.E.M., E.P.R., V.R., A.M., E.L., F.F., E.V., F.L.C., T.S. and F.A. designed experiments. P.E.M., E.P.R., E.L., V.R., A.M., F.F., T.S., F.A., J.B., E.V., V.M., M.T., and E.M. performed experiments and analyzed data. P.E.M. and E.P.R. wrote the paper with contributions from all authors.

## Additional information

**Competing interests:** P.E.M., T.O., and E.P.R. are named as inventors on a patent relating to the Nanoblades technology (patent applicants: Institut National de la Santé et de la Recherche Medicale (INSERM), Centre National de la Recherche Scientifique (CNRS), Ecole Normale Superieure de Lyon, Universite Claude Bernard Lyon 1, Villeurb-Anne Cedex; name of inventors: Theophile Ohlmann, Mathieu Misery, Philippe Mangeot, Emiliano Ricci; application number: WO 2017/068077 Al; patent status: published, 27th April 2017; all aspects of the manuscript are covered by the patent application. The remaining authors declare no competing interests.

