## [Peer Review File · Nature Communications]

Reviewers' Comments:

Reviewer #1:

Remarks to the Author:

This manuscript describes the development of MLV virus-like particles "nanoblades" to induce genome editing. The authors show that nanoblades can be packed with Cas9-sgRNA to generate indels in multiple cell lines as well as gene editing in vivo in mouse embryos and in adult animals. Nanoblades can also be complexed with donor templates to induce homology directed repair (HDR). This is an intriguing approach and a potential alternative to current genome editing methods, however, the manuscript is large in scope and fails to provide essential details for a thorough evaluation of the technique.

Comments:

1. The authors clearly show that nanoblades can be generated in HEK293T cells by transduction of multiple plasmids coding for a Gag-Cas9 fusion protein, sgRNA and other plasmids allowing the production of virus-like particles (VLPs). This concept, however, is not novel. The use of nanoblades as vehicles for cellular transduction -as mentioned by the authors- is based on previous studies that showed that retroviral Gag precursors can be used to deliver proteins to cells (Kaczmarczyk et al., PNAS, 2010; Voelkel et al., 2010). VLPs have also been used before to pack Cas9 protein and sgRNA (Qazi et al., Mol. Pharm. 2016). This somewhat diminishes the primacy of the manuscript.

2. The authors show that nanoblades can be used to generate indels in multiple cell types (HEK293T cells, hiPS cells, bone marrow cells, hematopoietic stem cells) and at three different loci. This is very commendable, nonetheless a better analysis is necessary to support their conclusions. For example, in figure 3a, the Western blot analysis needs an anti-DDX3 antibody assay alongside the anti-flag antibody analysis. Also, the quality of the Western blot is very poor and unconvincing. The PCR analysis lacks a description of the strategy used to amplify the region of the flag tag insertion and lacks information to document the flag tag insertion, i.e. to determine if it has been inserted in a proper orientation. In addition, there is a discrepancy between the PCR and western blot results. There seems to be no difference between the conditions of 0.1, 1, and 10 nmol/ml treatment in the PCR analysis which is not consistent with the Flag protein expression levels.

3. The use of nanoblades to modify zygotes requires microinjection into the perivitellin space. This does not offer a significant advantage over the traditional microinjection technique, since there is still a need to use sophisticated microinjection machinery and expertise. Also, even though the authors argue that injection into the perivitelline space is better for embryo viability, there is no data to support this argument. How many embryos were injected? How many made it to the two-cell stage? How many of the transferred embryos made it to birth? A table can be used to summarize this data. Also, what is the genetic background of the mice used in the experiments?

4. Nanoblades were used to inactivate the YFP gene in YFP transgenic embryos. These are poorly designed experiments. First, they are based on the inactivation of the YFP gene, which will lead to negative results (loss of YFP fluorescence). One can easily argue that non-fluorescent cells are the product of a different mutational event, for example, indirect targeting of the promoter region of the transgene rather than targeting the YFP gene itself. More concerning is the fact that the authors found that the YFP transgenic mice used in these experiments carry an average of 35 copies of the YFP gene. This negates the possibility of accurately assessing the indel frequency in the founder animals, and in consequence the effectiveness of nanoblades in vivo. The authors could have taken advantage of R26-YFP transgenic mice constitutively expressing YFP or even better, they could have targeted the Tyrosinase locus (to generate albino mice) and allow comparison with previous studies that targeted this locus using microinjection or electroporation

techniques (Yen et al., Dev. Biol. 2014; Chen et al, JBC. 2016).

5. Mosaicism is a common occurrence in gene editing experiments in mouse embryos (see Yen et al., Dev. Biol. 2014). The authors have not addressed this point at all.

6. In Figure 4a, mCherry fluorescence appears to be restricted to trophectodermal cells of blastocysts and there is no evidence that mCherry enters inner cell mass cells (the embryonic component of the conceptus).

7. There are significant residual cellular components in nanoblade particles (Supp. Fig. 2; Table1) that suggest heterogeneity in the content of the nanoblades. What are the effects of these components in their experiments?

8. It is not clear how this technology compares with current methods for gene editing. This needs to be added to the discussion section.

9. Image in Supplementary Figure 1b needs better description. Please provide arrows, dotted lines, etc. to describe the image. Also in panel 1d (right), what do "sgRNA(F+E)" means?.

10. Deep-sequencing procedure is missing in the Methods section as well as better description for other techniques.

Reviewer #2:

Remarks to the Author:

Genome editing is a highly enabling and promising capability for basic biological research, therapeutic development, and other applications. However, one significant hurdle is that genome editing enzymes are challenging to deliver, particularly to primary cells and in vivo. The most utilized approach for challenging cells in vitro and especially in vivo is viral-mediated delivery of cDNA encoding the enzyme, as well as the sgRNA in the case of Cas9. However, long-term expression of the Cas9 protein poses risks for off-target cutting and immune responses against the Cas9-expressing cells, so it is desirable to develop systems for delivery of transient Cas9 activity. This study fuses Cas9 to retroviral Gag, such that the enzyme (along with sgRNAs co-transfected into cells at the time of vehicle production) becomes incorporated into viral-like particles as they bud from the cell surface. Co-expression of the appropriate viral envelope protein yields VLPs that can enter cells in a receptor-mediated fashion and, following VLP fusion with the target cell membrane, deposit Cas9 inside the cell. The study is well-designed, the data are convincing, and the work is promising, at least for knockouts and oligo-based knock-ins. There are, however, several questions.

For clarity, can the authors provide a brief explanation about the previously published technique to incorporate DNA oligos into the VLPs? For Figure 3a, the Results mentions that the efficiency was oligo donor dose dependent. How was the amount of donor DNA varied (i.e. by adding more VLP, which would also change the amount of Cas9 delivered, or by changing the level of DNA loading into the VLPs). Also, can the authors estimate the % efficiency, for example by immunostaining cells for FLAG?

For the PuroR knock-in, can the authors describe more in the Results how the donor DNA was introduced into cells? The need to transfect donor DNA into cells in some ways negates the advantage of VLPs, as one can then simply also co-transfect mRNA encoding Cas9 or Cas9 RNPs. Also, what was the efficiency of colony generation, i.e. what was the number of colonies divided by the number of cells originally transfected? Six colonies does not sound like a high efficiency.

For the embryonic injections, why is the % chimerism relatively low (5-25%) despite injection as

early as the one cell stage? This is somewhat surprising, and it doesn't compare so favorably to other work involving introduction of Cas9 for zygotic editing that involve simpler electroporation rather than injection and that achieve higher efficiencies of editing in mouse (e.g. S. Chen et al., JBC 2016).

For the in vivo study targeting Hpd, it was reported that efficiency was 7-13% at two weeks. However, it is unclear whether this is the true efficiency, or whether positive selection for edited cells was occurring during this two weeks. Can the authors clarify?

In sum, this is an interesting and versatile technology, though one with some limitations. Addressing a number of questions would benefit the manuscript.

Point-by-point rebuttal:

Reviewer #1:

This manuscript describes the development of MLV virus-like particles “nanoblades” to induce genome editing. The authors show that nanoblades can be packed with Cas9-sgRNA to generate indels in multiple cell lines as well as gene editing *in vivo* in mouse embryos and in adult animals. Nanoblades can also be complexed with donor templates to induce homology directed repair (HDR). This is an intriguing approach and a potential alternative to current genome editing methods, however, the manuscript is large in scope and fails to provide essential details for a thorough evaluation of the technique.

We thank the reviewer for highlighting the potential of Nanoblades as an alternative to other genome editing protocols available in the literature. The main goal of our study was to provide an overview of all the possible applications that Nanoblades could have in different experimental models and setups, ranging from genome editing in immortalized cells and difficult to transfect primary cells, up to mouse embryos and adult mice. We also explored potential applications with orthogonal Cas9 to induce gene-specific expression so that readers could have a glimpse of all the other possible applications of Nanoblades and explore them to further develop the technology. As a consequence, the manuscript is indeed large in scope and some of the showcased applications could be further optimized to achieve better results. However, since Nanoblades are easy to produce in any laboratory equipped with a tissue culture facility, we believe that our results will attract readers to test Nanoblades and optimize their use in their favorite model.

We expect that the new results provided in this revision will provide the reviewer and readers with the required details to evaluate the technique.

Comments:

1. The authors clearly show that nanoblades can be generated in HEK293T cells by transduction of multiple plasmids coding for a Gag-Cas9 fusion protein, sgRNA and other plasmids allowing the production of virus-like particles (VLPs). This concept, however, is not novel. The use of nanoblades as vehicles for cellular transduction -as mentioned by the authors- is based on previous studies that showed that retroviral Gag precursors can be used to deliver proteins to cells (Kaczmarczyk et al., PNAS, 2010; Voelkel et al., 2010). VLPs have also been used before to pack Cas9 protein and sgRNA (Qazi et al., Mol. Pharm. 2016). This somewhat diminishes the primacy of the manuscript.

Our approach was indeed inspired from two previous studies showing that proteins can be fused to Gag precursors in order to deliver them to recipient cells. However, in our case we show, for the first time, that we can package a protein (Cas9) and its bound RNA (sgRNA) to mediate genome-editing in primary cells and *in vivo*. Furthermore, we use a novel hybrid pseudotyping approach combining VSVg and a Baboon endogenous retrovirus engineered envelope (BRL), which is essential for the efficiency of Cas9 delivery (See Supplementary Figure 6) and has not been previously described in the literature.

Although the VLPs described by Qazi et al., Mol Pharm. 2016 can be conceptually similar to our approach (Cas9 fusion to a viral protein), their strategy was performed in bacteria using a protein from a bacteriophage that is fused to Cas9. Importantly, the authors do not show any functional delivery of their VLP cargo in any kind of cells and their functional

assays are limited to *in vitro* experiments were the content of their Cas9-loaded VLPs is incubated with purified linear or circular plasmids. We therefore do not think that this report can be compared to our Nanoblades approach.

2. The authors show that nanoblades can be used to generate indels in multiple cell types (HEK293T cells, hiPS cells, bone marrow cells, hematopoietic stem cells) and at three different loci. This is very commendable, nonetheless a better analysis is necessary to support their conclusions. For example, in figure 3a, the Western blot analysis needs an anti-DDX3 antibody assay alongside the anti-flag antibody analysis. Also, the quality of the Western blot is very poor and unconvincing. The PCR analysis lacks a description of the strategy used to amplify the region of the flag tag insertion and lacks information to document the flag tag insertion, i.e. to determine if it has been inserted in a proper orientation. In addition, there is a discrepancy between the PCR and western blot results. There seems to be no difference between the conditions of 0.1, 1, and 10 nmol/ml treatment in the PCR analysis which is not consistent with the Flag protein expression levels.

We agree with the reviewer that the quality of the western blot presented in Figure 3a was not convincing to support such an important result. We have therefore repeated the experiment as described below.

Nanoblades targeting DDX3 were complexed with ssDNA donor repair template bearing the Flag sequence and homology arms to the DDX3 locus in presence of polybrene. These complexed Nanoblades were incubated with HEK293T for 24 hours, and cells were grown and lysed to recover proteins and genomic DNA. An anti-Flag immunoprecipitation was performed on the cell lysate followed by specific elution with a Flag-peptide. The elution products were then analysed by western-blotting using anti-Flag and anti-DDX3 antibodies. Our results (presented in Figure 3a) now clearly show that cells transduced with these “All-in-one” Nanoblades express a Flag-tag DDX3 protein from the endogenous locus as we could detect signal at the expected size both with the anti-DDX3 and anti-flag antibodies. We further validated these results using two different PCR based assays which confirmed 1. the insertion of the flag sequence (using oligos flanking the AUG codon of DDX3, outside of the recombination arms of the ssDNA donor template); 2. its correct orientation (using one oligo that binds to the Flag sequence in the forward orientation and a reverse oligo that binds downstream of the AUG codon of DDX3). The scheme for each PCR strategy now appears in Figure 3a so that it is clear for the reader how we assessed the insertion of the flag sequence and its orientation.

To further assess the efficiency of Flag-recombination, we derived single-cell clones from the bulk population of HEK293T cells transduced with “All-in-one” Nanoblades and we tested for the insertion of the Flag-sequence using the flanking PCR assay. Our results show that 12 out of the 20 clones that we tested were positive for the Flag-insertion thus suggesting a recombination rate of at least 50% of transduced cells. The western-blot analysis and the PCR assays in the bulk population were performed in three independent replicates all showing similar results. However, the efficiency of recombination was not always correlated to the amount of ssDNA donor repair template added to Nanoblades. We therefore removed this information from the text.

We thank the reviewer for the critical comments regarding this figure, which helped us to obtain a more convincing figure.

3. The use of nanoblades to modify zygotes requires microinjection into the perivitellin space. This does not offer a significant advantage over the traditional microinjection technique, since there is still a need to use sophisticated microinjection machinery and expertise. Also, even though the authors argue that injection into the perivitelline space is

better for embryo viability, there is no data to support this argument. How many embryos were injected? How many made it to the two-cell stage? How many of the transferred embryos made it to birth? A table can be used to summarize this data. Also, what is the genetic background of the mice used in the experiments?

All the experiments related to zygotes were performed by the personnel of our local animal facility. Based on their experience with Nanoblades, although there is still a requirement for a microinjection machinery, the injection gesture is easier and much more rapid to perform than traditional microinjection in the nucleus of oocytes.

To support our claims regarding the viability of embryos injected with Nanoblades, we have now included a table with embryo viability numbers (Figure 4c) both at the “two cells” and “blastocyst” stages. Our results show no significant differences between non-injected and Nanoblades-injected embryos (117 out of 117 (100%) embryos injected with Nanoblades made it to the two-cells stage and 57 out of 60 (94.7%) made it from the two-cell to the blastocyst stage). We also include the number of implanted embryos (injected with Nanoblades) that made it to birth (roughly 30%), although in this case we do not have numbers for non-injected embryos.

4. Nanoblades were used to inactivate the YFP gene in YFP transgenic embryos. These are poorly designed experiments. First, they are based on the inactivation of the YFP gene, which will lead to negative results (loss of YFP fluorescence). One can easily argue that non-fluorescent cells are the product of a different mutational event, for example, indirect targeting of the promoter region of the transgene rather than targeting the YFP gene itself. More concerning is the fact that the authors found that the YFP transgenic mice used in these experiments carry an average of 35 copies of the YFP gene. This negates the possibility of accurately assessing the indel frequency in the founder animals, and in consequence the effectiveness of nanoblades in vivo. The authors could have taken advantage of R26-YFP transgenic mice constitutively expressing YFP or even better, they could have targeted the Tyrosinase locus (to generate albino mice) and allow comparison with previous studies that targeted this locus using microinjection or electroporation techniques (Yen et al., Dev. Biol. 2014; Chen et al, JBC. 2016).

Experiments carried out on YFP transgenic embryos can indeed be criticized regarding the phenotype reading and the transgene copy number. However, they can still be valuable to monitor gene editing (as this was done by genetic test using the T7 endonuclease assay) and transmission to the F1 generation (which we clearly show for the YFP mice, see Supplementary Figure 5a). To overcome the issue related to the reading of the phenotype and the gene copy number, we further made use of the ROSA LoxP-Stop-LoxP-YFP mice, which have a single copy of the YFP under control of a LoxP cassette with polyadenylation signals to block YFP expression. In this case, we showed that Nanoblades programmed with a sgRNA targeting the LoxP site were able to induce YFP expression in the F0 (in most tissues) and that the editing was transmitted to the F1 generation (see Supplementary Figure 5b).

However, we agree with the reviewer that targeting an endogenous gene might be more relevant than a transgene. To this aim, we decided to use the sgRNA targeting the Tyrosinase locus described in Figure 1 of Chen et al., JBC. 2016 as suggested by the reviewer. We prepared Nanoblades programmed with this sgRNA and we microinjected them in the perivitelline space of BL57/CJ6 embryos that we cultured until the blastocyst stage. The results (presented in Figure 4) show that 16 out of 40 blastocysts tested were positive for genome editing, 2 of them showing complete editing at all alleles. As a

consequence of this new addition, results obtained in the YFP and ROSA LoxP-Stop-LoxP-YFP mice are now presented as Supplementary Figure 5.

We hope that all together, this new data combined with the previous YFP experiments, will be sufficient to show that Nanoblades can be effectively used to modify zygotes and transmit the modification to the offspring.

5. Mosaicism is a common occurrence in gene editing experiments in mouse embryos (see Yen et al., Dev. Biol. 2014). The authors have not addressed this point at all.

In the ROSA LoxP-Stop-LoxP-YFP experiments (Supplementary Figure 5b, bottom panels), the F0 mouse obtained upon Nanoblades injection showed YFP expression in most tissues and in a uniform pattern. This was further confirmed by PCR amplification of the targeted cassette from a biopsy, which indicates that the excision of the LoxP cassette was very efficient, although not complete, thus arguing for some mosaicism (Supplementary Figure 5b, bottom left panel).

In our new results with the sgRNA targeting the *Tyrosinase* locus, we obtained at least three blastocysts (out of 40 tested) with an apparent complete editing of the Tyr locus in all cells of the embryo. In the remaining 13 blastocysts that were positive for genome editing, we observed different editing efficiencies thus arguing for variable levels of mosaicism between individuals.

We have now introduced a sentence in the results section stating that mosaicism is present in variable levels in F0 animals but that in most cases, the edited loci are transmitted to the F1 generation.

6. In Figure 4a, mCherry fluorescence appears to be restricted to trophectodermal cells of blastocysts and there is no evidence that mCherry enters inner cell mass cells (the embryonic component of the conceptus).

We agree with the reviewer that most of the mCherry signal appears to be located in trophectodermal cells. Nevertheless, we obtain genetically modified individuals at the F0. We also have data using Nanoblades loaded with the Cre protein injected in the perivitelline space of ROSA LoxP-Stop-LoxP-YFP embryos leading to 96% of F0 newborn mice (24 out of 25 mice) displaying YFP expression (this data is not included in the manuscript but we can share if requested by reviewers). We are therefore confident that Nanoblades do enter the inner cell mass (at some point) to deliver their content. It is possible that most of the signal is enriched in trophectodermal cells thus blinding the signal present in the inner cell mass. If the reviewer think it is better to remove these pictures for clarity reasons, we will understand the arguments for doing so.

7. There are significant residual cellular components in nanoblade particles (Supp. Fig. 2; Table1) that suggest heterogeneity in the content of the nanoblades. What are the effects of these components in their experiments?

This is a very relevant question both regarding Nanoblades and other viral delivery vectors such as lentiviral vectors, which have also been shown to carry cellular proteins. To try to measure the impact of residual cellular components carried by Nanoblades into recipient cells, we have tested two different technical approaches described below.

First, we over-expressed the Firefly-luciferase protein (cytosolic protein) in producer cells and collected Nanoblades from the supernatant. We then transduced HEK293 cells for 24

hours, washed them in PBS twice, added fresh medium and measured luciferase activity at different time points after washing. Our results (presented in Supplementary Figure 2c) indicate that luciferase delivery is limited in amount and rapidly cleared in recipient cells (almost undetectable 48h after transduction).

To further validate these results in a more physiological context, we followed the delivery of an endogenous cellular protein (CD81) that is abundant at the plasma membrane of producer cells and in Nanoblades as detected by Mass-spectrometry. HepG2 cells (a cell-line that lacks CD81 expression as shown in Zhang J et al., J Virol. 2004) were transduced with Nanoblades for 24 hours and then cells were washed twice with PBS. After the washings, CD81 levels were monitored in HepG2 cells by FACS analysis at 0, 8 and 48 hours. Results (presented in Supplementary Figure 2d) clearly show that while CD81 is very abundant in HEK293T producer cells and completely absent in HepG2 cells, transduction with Nanoblades mildly increase CD81 levels in recipient cells and this was detectable only immediately after the PBS washing but no-longer at 8 or 48 hours after transduction.

We conclude that the impact of residual cellular components delivered by Nanoblades is limited both in time and amount and therefore should not significantly impact the physiology of transduced cells.

8. It is not clear how this technology compares with current methods for gene editing. This needs to be added to the discussion section.

In Figure 1b and Supplementary Figure 1e we compared Nanoblades to lipofection and electroporation in U2OS cells and primary human fibroblasts and, in both cases, Nanoblades outperformed the other two delivery methods in terms of efficiency and speed of gene editing. Nevertheless, we agree with the reviewer that the discussion lacked a paragraph contrasting our technique with other available methods. We have now added this information in the discussion section to compare Nanoblades with other techniques in cultured cells, primary cells and *in vivo*.

9. Image in Supplementary Figure 1b needs better description. Please provide arrows, dotted lines, etc. to describe the image. Also in panel 1d (right), what do “sgRNA(F+E)” means?.

We thank the reviewer for the suggestions to improve the figures. We now added better descriptions of the electron microscopy pictures. We apologize for the missing information regarding the sgRNA(F+E). It corresponds to an optimized sgRNA described in Chen B et al., Cell. 2013 that is supposed to improve binding to Cas9. However, in our experiments we did not detect any improvement regarding its incorporation within Nanoblades compared to the non optimized version of the sgRNA. This information is now clearly described in the results section and the figure legend.

10. Deep-sequencing procedure is missing in the Methods section as well as better description for other techniques.

We apologize for the missing information, that is now included in the methods section.

Reviewer #2:

Genome editing is a highly enabling and promising capability for basic biological research, therapeutic development, and other applications. However, one significant hurdle is that genome editing enzymes are challenging to deliver, particularly to primary cells and in vivo. The most utilized approach for challenging cells in vitro and especially in vivo is viral-mediated delivery of cDNA encoding the enzyme, as well as the sgRNA in the case of Cas9. However, long-term expression of the Cas9 protein poses risks for off-target cutting and immune responses against the Cas9-expressing cells, so it is desirable to develop systems for delivery of transient Cas9 activity. This study fuses Cas9 to retroviral Gag, such that the enzyme (along with sgRNAs co-transfected into cells at the time of vehicle production) becomes incorporated into viral-like particles as they bud from the cell surface. Co-expression of the appropriate viral envelope protein yields VLPs that can enter cells in a receptor-mediated fashion and, following VLP fusion with the target cell membrane, deposit Cas9 inside the cell. The study is well-designed, the data are convincing, and the work is promising, at least for knockouts and oligo-based knock-ins. There are, however, several questions.

For clarity, can the authors provide a brief explanation about the previously published technique to incorporate DNA oligos into the VLPs? For Figure 3a, the Results mentions that the efficiency was oligo donor dose dependent. How was the amount of donor DNA varied (i.e. by adding more VLP, which would also change the amount of Cas9 delivered, or by changing the level of DNA loading into the VLPs). Also, can the authors estimate the % efficiency, for example by immunostaining cells for FLAG?

We thank the reviewer for the overall positive comments regarding our work.

As suggested, we have now added a brief explanation in the results section describing the previous use of polybrene to complex DNA and viral particles for efficient transfection of the donor DNA. We have also repeated the “All-in-one” Flag-DDX3 knock-in experiment to obtain more convincing results for the western-blot and genotyping PCR assays. In this experiment, we tested again different doses of donor oligomers while the amount of Nanoblades remained constant (only the amount of donor DNA was increased). After repeating the experiments in three independent biological replicates, the efficiency of recombination was not always correlated to the amount of ssDNA donor repair template added to Nanoblades. We therefore removed this information from the text.

In order to estimate the efficiency of recombination, as suggested by the reviewer, we performed limit dilution experiments on the bulk population of cells transduced with “All-in-one” Flag-DDX3 Nanoblades and obtained single-cell clones that we expanded and tested for insertion of the Flag sequence at the DDX3 locus. Our results (Figure 3a bottom panel) show that 12 out of the 20 clones tested were positive for the Flag-insertion, thus suggesting a recombination rate of at least 50% of treated cells.

For the PuroR knock-in, can the authors describe more in the Results how the donor DNA was introduced into cells? The need to transfect donor DNA into cells in some ways negates the advantage of VLPs, as one can then simply also co-transfect mRNA encoding Cas9 or Cas9 RNPs. Also, what was the efficiency of colony generation, i.e. what was the number of colonies divided by the number of cells originally transfected? Six colonies does not sound like a high efficiency.

We have now added a more precise description of the PuroR knock-in experiment in the results section. This experiment was performed similarly to the Flag-DDX3 knock-in experiment by complexing Nanoblades with the donor dsDNA template and polybrene in

the absence of any transfection reagent. We unfortunately do not have the information regarding the ratio of transduced cells and puromycin resistant clones but we did obtain more than 6 colonies. However we only picked 6 colonies as we were only looking to validate the correct insertion of the transgene in the AAVS1 locus.

For the embryonic injections, why is the % chimerism relatively low (5-25%) despite injection as early as the one cell stage? This is somewhat surprising, and it doesn't compare so favorably to other work involving introduction of Cas9 for zygotic editing that involve simpler electroporation rather than injection and that achieve higher efficiencies of editing in mouse (e.g. S. Chen et al., JBC 2016).

The relative low chimerism observed in the YFP mice can be explained by the fact that these mice carried close to 35 copies of the YFP transgene per cell and therefore not all alleles were edited in each cell. However, the high rate of transmission of the edited loci to the F1 generation argues favorably for an efficient editing in most cells of the F0 mice.

In order to obtain additional results with an endogenous gene, we have repeated the embryonic injections using the sgRNA that targets the *Tyrosinase* gene described in S Chen et al., JBC. 2016. The new results obtained at the blastocyst stage are presented in Figure 4d. Out of 40 tested blastocysts, 16 appeared to be edited at the *Tyr* locus and 3 of them displayed what appears to be complete editing in all cells (Figure 4d see "HinfI restriction" panel). In the remaining 13 blastocysts that were positive for genome editing, we observed different editing efficiencies thus arguing for variable levels of mosaicism between individuals. We have therefore now introduced a sentence in the results section stating that mosaicism is present in variable levels in F0 animals but that in most cases, the edited loci are transmitted to the F1 generation.

Although in this setup Nanoblades are slightly less efficient than electroporation, they do not affect embryo viability which could represent an advantage. We also think that there is room for improving the efficiency of Nanoblades in generating transgenic embryos (for example by optimising the purification and concentration of injected Nanoblades). However, we think this optimisation work goes beyond the scope of the current study, which mainly focus is to showcase the potential applications of VLP-mediated delivery of the Cas9/sgRNA complex.

For the in vivo study targeting Hpd, it was reported that efficiency was 7-13% at two weeks. However, it is unclear whether this is the true efficiency, or whether positive selection for edited cells was occurring during this two weeks. Can the authors clarify?

We did not test the efficiency of editing sooner than two weeks post-infection. Although positive selection might have occurred during these two weeks, we do not expect it to be very important since we only observed a mild positive selection between two and four weeks post-infection. This is now commented in the results section so that it is clear for readers that positive selection could have occurred and affected the observed efficiency at two-weeks.

In sum, this is an interesting and versatile technology, though one with some limitations. Addressing a number of questions would benefit the manuscript.

We thank the reviewer for her/his comments and hope that the revised version of the manuscript addresses her/his main concerns.

Reviewers' Comments:

Reviewer #1:

Remarks to the Author:

The authors have satisfactorily answered most of my previous concerns, however, there is still some major points and a couple of corrections that need to be addressed.

Comments:

1. The authors have heeded a previous suggestion to test nanoblades at the Tyrosinase locus (an endogenous loci). After T7 Endonuclease assays, they have found good evidence for indels at this locus (40%) in blastocysts. These results, however, are not sufficient to validate the effectiveness of nanoblades for gene editing *in vivo*. The authors need to test the efficiency of nanoblades by transferring the blastocysts (or earlier stage embryos) into pseudopregnant females and allowing them to develop to term. These experiments will allow the assessment of mosaicism in pups and a comparison of the *in vivo* efficiency of nanoblades relative to other technologies. Targeting Tyrosinase is particularly helpful since it has been used to test gene editing by pronuclear injection (Yen et al., 2014. *Dev. Biol.*), electroporation (Chen et al., 2016. *J. Biol. Chem.*) and the recently published AAV-Crispr-Cas9 technique (Yoon et al., 2018. *Nat. Comm.*).

2. Using the T7 endonuclease assay, the authors provide evidence that mosaicism is present in Tyrosinase gene-edited embryos but there is no sequence information about the nature of the mutations or the number of mutations present per individual.

3. In the previous critique it was pointed out that it was not possible to discern the location of the Cherry VLPs in the blastocyst picture in figure 4a. This picture and the one of the 2-cell stage embryo are good illustrations supporting the nanoblades technology, however, DIC pictures should be added and a merge of the DIC and fluorescence pictures should be made to clearly show the location of the VLPs.

Minor comments:

The cartoon in Figure 4a refers to a fertilized oocyte in which case it should be called "mouse zygote" not "mouse oocyte". Similarly in figure 4c the correct name should be "2-cell embryos" not "2 cells embryos"

The legend of figure 4 needs to be corrected. The pictures of embryos are not the bottom panels of fig. 4a.

Figure 4c is misleading. The top two tables refer to gene editing of the Tyrosinase locus but the bottom table refers to a different experiment. There was no transfer or newborns obtained from Tyrosinase-edited embryos.

The C57BL/6J strain name is misspelled several times in the manuscript as BL57/CJ6.

Reviewer #2:

Remarks to the Author:

This reviewer thanks the authors for addressing several of the comments. However, for the applications pursued, there are competing approaches. That is, for larger cassette knock-in's *in vitro*, there is electroporation of RNPs plus plasmids, or RNP electroporation plus infection with AAV carrying the donor. For zygotic editing, there is electroporation with RNPs and oligos, or more recently use of AAV (work by Gao). As a result, while this approach is novel, it's not sufficient to

present a novel approach without benchmarking it quantitatively compared to other methods (the reason that this reviewer asked numerous questions about efficiency in the prior review). Quantitative comparison should include not only % of transduced/transfected/infected cells that become gene modified, but some indication of what % are correctly targeted. Saying that it was more than six colonies (in the case of purymycin knock-in) isn't necessarily convincing that this is an efficient approach relative to well-established ones.

For oligo knock-in's, there is potentially some advantage here in viability. That is, the cell viability following infecting with a viral particle complexed via polybrene with oligos might be better than corresponding viability of electroporating a RNP (or mRNA) plus donor oligo. However, the argument should be made quantitatively, where both viability and efficiency (% overall cells, and % clones with correct on-target) are benchmarked/compared to the best current method.

The in vivo genome editing is potentially very interesting. However, how does this efficiency compare to that with published work with lipid nanoparticles (which report 80% editing, e.g. Yin et al., Nat Biotech, 2017).

In sum, this method is halfway between a viral and a non-viral approach. It is not likely to be as efficient as a viral approach, but one can make the reasonable argument that one is willing to sacrifice on efficiency in order to get other benefits associated with transient activity. In that case, however, the method had better compare well to the non-viral approaches that have this advantage. Since there is sparse comparison or benchmarking, it is not clear the method meets either bar.

Point-by-point rebuttal:

Reviewer #1:

The authors have satisfactorily answered most of my previous concerns, however, there is still some major points and a couple of corrections that need to be addressed.

Comments:

1. The authors have heeded a previous suggestion to test nanoblades at the Tyrosinase locus (an endogenous loci). After T7 Endonuclease assays, they have found good evidence for indels at this locus (40%) in blastocysts. These results, however, are not sufficient to validate the effectiveness of nanoblades for gene editing in vivo. The authors need to test the efficiency of nanoblades by transferring the blastocysts (or earlier stage embryos) into pseudopregnant females and allowing them to develop to term. These experiments will allow the assessment of mosaicism in pups and a comparison of the in vivo efficiency of nanoblades relative to other technologies. Targeting Tyrosinase is particularly helpful since it has been used to test gene editing by pronuclear injection (Yen et al., 2014. Dev. Biol.), electroporation (Chen et al., 2016. J. Biol. Chem.) and the recently published AAV-Crispr-Cas9 technique (Yoon et al., 2018. Nat. Comm.).

We are glad that our first revision addressed most of the referee's concerns and we agree that additional evidence supporting the use of Nanoblades in generating transgenic mice will significantly strengthen our manuscript. To this aim, we have repeated perivitelline injection of Nanoblades in mouse embryos using Nanoblades programmed with two sgRNAs targeting the *Tyrosinase* locus, one from the recently published AAV-CRISPR-Cas9 (Yoon et al., 2018 Nat. Comm) and a second sgRNA that we designed to target the *Tyr* open-reading 180bp upstream the first one. Injected embryos were then transferred into pseudopregnant females in order to obtain F0 mice. The results of this experiment are now presented in Figure 4e. To summarize our results, 5 out of 8 F0 mice have visible signs of genome editing at the *Tyr* locus. Among these, 2 displayed a complete albino phenotype, 3 displayed partial disruption of the *Tyr* gene and 3 displayed a wild-type phenotype.

2. Using the T7 endonuclease assay, the authors provide evidence that mosaicism is present in Tyrosinase gene-edited embryos but there is no sequence information about the nature of the mutations or the number of mutations present per individual.

The F0 mice were genotyped by extracting genomic DNA from their fingers and amplifying the targeted *Tyr* locus by PCR. The efficiency of genome-editing was then assessed by Sanger sequencing of the bulk PCR products and performing TIDE analysis of the sequencing electropherograms. We also cloned individual PCR products to characterize some of the indels generated. One of the two fully albino mice had a complete bi-allelic deletion of the *Tyr* genomic sequence located between the two targeted sites. This deletion was found in all cells as tested by Sanger sequencing of a bulk PCR amplicon prepared from genomic DNA extracted from the mouse's fingers (See the electropherogram in Figure 4e bottom panel). This result was confirmed by sequencing 10 individual PCR clones obtained from TOPO-cloning of the PCR amplicon, which all showed an identical *Tyr* sequence lacking the segment between the two targeted loci (data not shown). In the remaining mice, we detected editing efficiencies ranging from 11 to 78% of all *Tyr* alleles (as assessed from genomic DNA extracted from the fingers; Figure 4e). Interestingly, in these mice sgRNA1 (corresponding to the sgRNA used by Yoon et al., 2018) appeared more efficient in inducing INDELS than sgRNA2 (See Figure 4e genomic

alignment panel). Nevertheless, the degree of mosaicism detected in each mouse appeared very limited and similar to that observed in other studies (Chen et al., 2016 JBC and Yoon et al., 2018 Nat Commun). For example, we could only detect two different types of INDELS within mice #7 and #8 upon sequencing of 10 different clones for each of them. This suggests that Nanoblades editing occurred at early stages of embryo development and for a limited time window.

3. In the previous critique it was pointed out that it was not possible to discern the location of the Cherry VLPs in the blastocyst picture in figure 4a. This picture and the one of the 2-cell stage embryo are good illustrations supporting the nanoblades technology, however, DIC pictures should be added and a merge of the DIC and fluorescence pictures should be made to clearly show the location of the VLPs.

We agree with the reviewer that having DIC pictures would help to clearly show the location of the VLPs in the blastocysts and we were planning on obtaining such pictures on time for the resubmission. However, due to a misunderstanding with our local animal facility, we were not able to generate those new data on time (an injection experiment is now scheduled for mid October). Since we believe that such pictures are not essential to assess the efficiency of Nanoblades in generating transgenic mice, we decided to resubmit the manuscript before obtaining the pictures. We will nevertheless introduce the results as soon as we obtain them. We hope the reviewer will understand our motivations for such a choice.

Minor comments:

The cartoon in Figure 4a refers to a fertilized oocyte in which case it should be called "mouse zygote" not "mouse oocyte". Similarly in figure 4c the correct name should be "2-cell embryos" not "2 cells embryos"

We thank the reviewer for pointing out these errors, we have now corrected them as suggested.

The legend of figure 4 needs to be corrected. The pictures of embryos are not the bottom panels of fig. 4a.

The legend has now been corrected accordingly.

Figure 4c is misleading. The top two tables refer to gene editing of the Tyrosinase locus but the bottom table refers to a different experiment. There was no transfer or newborns obtained from Tyrosinase-edited embryos.

Indeed, the bottom table refers to the experiments that are now presented in Supplementary figure 5. The legend of the figure has therefore been corrected accordingly.

The C57BL/6J strain name is misspelled several times in the manuscript as BL57/CJ6. The misspelling has now been corrected throughout the manuscript.

Reviewer #2:

This reviewer thanks the authors for addressing several of the comments. However, for the applications pursued, there are competing approaches. That is, for larger cassette knock-in's in vitro, there is electroporation of RNPs plus plasmids, or RNP electroporation plus infection with AAV carrying the donor. For zygotic editing, there is electroporation with

RNPs and oligos, or more recently use of AAV (work by Gao). As a result, while this approach is novel, it's not sufficient to present a novel approach without benchmarking it quantitatively compared to other methods (the reason that this reviewer asked numerous questions about efficiency in the prior review). Quantitative comparison should include not only % of transduced/transfected/infected cells that become gene modified, but some indication of what % are correctly targeted. Saying that it was more than six colonies (in the case of puromycin knock-in) isn't necessarily convincing that this is an efficient approach relative to well-established ones. For oligo knock-in's, there is potentially some advantage here in viability. That is, the cell viability following infecting with a viral particle complexed via polybrene with oligos might be better than corresponding viability of electroporating a RNP (or mRNA) plus donor oligo. However, the argument should be made quantitatively, where both viability and efficiency (% overall cells, and % clones with correct on-target) are benchmarked/compared to the best current method.

The *in vivo* genome editing is potentially very interesting. However, how does this efficiency compare to that with published work with lipid nanoparticles (which report 80% editing, e.g. Yin et al., Nat Biotech, 2017).

We agree with the reviewer that the CRISPR/Cas9 field is highly competitive and that several methods have been recently developed to mediate efficient genome editing *in vivo* and in cultured cells. However, each method has its own set of benefits and drawbacks. For example, protein electroporation is relatively expensive, requires specific reagents and material (electroporator) which makes it not quite accessible or cost-effective for all users, particularly when large amounts of cells are required. Furthermore, it can be associated with cell mortality, especially when working with primary cells.

AAV vectors are very efficient, particularly *in vivo*, and they are supported by years of continuous efforts to improve their packaging capacity and transduction efficiency. However they do not deliver the active protein directly and thus there is a time gap between transduction and Cas9 and sgRNA expression thus delaying genome editing. Furthermore, AAV vectors express the Cas9 for a longer time frame than protein electroporation thus increasing the chances of generating off-target effects. Similarly, lentiviral vectors also lead to a Gap in Cas9/sgRNA expression and there can be drawbacks associated with the integration of the expression cassette into the host genome. Some of these drawbacks can be avoided using non-integrating lentiviral vectors but their expression levels of Cas9 are not always sufficient to mediate efficient genome-editing.

Our Nanoblades approach is relatively inexpensive (we have estimated its cost to 40 euros, including consumables and labour, for a dose enough to transduce at least 2 million cultured cells and its upscaling costs are degressive) and does not require any specific material other than those found in a regular cell culture laboratory. Their preparation protocol is very simple and they can be stored for up to a month at 4C degrees. Cas9 delivery is rapid and efficient in cultured cells and primary cells that are difficult to transfect/transduce. This is particularly important in primary cells which usually have a limited life-time. Moreover, we have shown that Nanoblades can easily incorporate Cas9 variants to perform other applications not related to genome-editing thus being more versatile than recombinant protein approaches that require purification of the protein from bacteria or other viral based vectors that could be limited by their packaging capacity. We therefore strongly believe that Nanoblades could represent a simple and easy to adopt tool for most academic laboratories looking to perform genome-editing in their favorite cell type. We have many collaborators currently using Nanoblades to inactivate gene expression in primary cells or to induce dsDNA breaks in order to study DNA repair pathways in primary cells derived from patients and they all appreciate how simple the transduction protocol is compared to other delivery methods.

In our previous version of the manuscript, we had compared Nanoblades efficiency to DNA transfection and DNA electroporation in immortalized cell lines and in primary fibroblasts (Figure 1b and Supplementary Figure 1e). In these case scenarios, Nanoblades outperformed DNA transfection and electroporation both in terms of rapidity of induction of dsDNA breaks and overall efficiency. We also showed that Nanoblades lead to less off-target effects than traditional DNA transfection. We nevertheless agree with the reviewer that a comparison of Nanoblades with other protein delivery protocols would be important to illustrate the benefits of our approach. To this aim, we have now incorporated a new panel in Figure 2 (see Figure 2c) where we have compared Nanoblades to Cas9/sgRNA RNP electroporation in HEK293T cells and in primary mouse bone marrow cells, which are known to be difficult to transfect/transduce. For this, we obtained the commercially available EnGen Cas9 NLS from New England Biolabs (M0646T) that we complexed with *in vitro* transcribed sgRNAs that were prepared using the EnGen sgRNA synthesis kit from New England Biolabs (E3322). These complexes were then electroporated into cells using the Neon Electroporation system (Thermo Fisher Scientific) as specified by the EnGen Cas9 protocol. In HEK293T cells we used the settings suggested by NEB (1700V, 20ms, 1pulse), while for primary bone marrow cells we used two different protocols, one for mouse immortalized macrophages (1680V, 20ms, 1 pulse) and one for mouse M1 myeloid suspension cells (1350V, 10ms, 4 pulses). In parallel, we transduced cells with Nanoblades. In HEK293T cells, we used an sgRNA targeting *EMX1* while in mouse bone marrow cells we used an sgRNA targeting the *Fto* gene. 24 hours after electroporation/transduction, we extracted genomic DNA and monitored genome-editing by T7 endonuclease assay and TIDE analysis of the Sanger sequencing electropherograms. Our results indicate that Nanoblades and protein electroporation perform relatively similarly in HEK293T cells (71% of indels in the Nanoblades condition Vs 44% for Cas9/sgRNA electroporation). However, in primary bone marrow cells, Nanoblades largely outperform protein electroporation as shown by the T7 endonuclease assay (Figure 2c). In these cells, T7 endonuclease assays showed that Nanoblades were able to edit most of the *Fto* alleles while Cas9/sgRNA electroporation only showed a mild editing signal. Strikingly, TIDE analysis of the Sanger sequencing electropherograms indicated that Nanoblades were able to edit up to 76% of the *Fto* alleles while protein electroporation did not show any detectable editing activity (probably as a consequence of the editing being below the detection level of the TIDE software). With both methods we observed a very mild effect on cell viability 24h after Cas9 delivery as measured by propidium iodide staining (Supplementary figure 2e). We are therefore confident that Nanoblades can be a viable and democratic alternative to other well-established methods. We cannot exclude that optimization of the electroporation conditions could improve Cas9 delivery in bone marrow cells, but this again argues in favor of Nanoblades, which with a simple and somewhat “universal” transduction protocol, appear to work efficiently in most of the situations that we have tested.

Regarding the knock-in experiments with dsDNA donors, we agree with the reviewer that our previous figure was not clear with respect to the overall efficiency. We have therefore repeated the experiment by transducing 1×10^5 HEK293T cells with Nanoblades complexed with the same 4kb dsDNA puromycin cassette in the presence of polybrene. After puromycin selection, we obtained a total of 47 visible colonies that we screened by PCR in order to validate the correct insertion of the cassette at the AAVS1 locus. 42 out of the 47 clones had the puromycin cassette correctly integrated at the expected locus. Because there is clearly room for improvement regarding the use of large dsDNA donors, we are currently exploring alternative routes for delivering the donor DNA inside the VLPs. However these improvements will require much more work before yielding any significant results and we thus believe they are out of the scope of this manuscript.

We have also repeated the knock-in experiment using the donor DNA oligo to measured cell viability by propidium iodide incorporation. The results, presented in Supplementary figure 3a) show that Nanoblades do not have a significant impact on cell viability when performing knock-in experiments.

Finally, regarding the use of Nanoblades *in vivo*, although currently Nanoblades do not outperform Cas9/sgRNA electroporation in mouse embryos, they are still very competitive in terms of efficiency (See the new version of figure 4e). Moreover, they are not associated with significant embryo mortality and thus could represent a viable, simple to implement, alternative. Similarly, the *in vivo* editing efficiency in the liver of mice upon Nanoblades is significantly lower than that recently reported using lipid nanoparticles and chemically modified sgRNAs (Yin et al., 2017 Nat Biotech). Nevertheless, we could achieve 10% of genome editing in the liver without any particular optimization of the protocol and simply by producing a batch of VLPs in a BSL2 laboratory. The focus of our work was not to specifically optimize our VLPs for *in vivo* editing of hepatocytes but rather to showcase the many potentials of our method in various experimental setups. We are convinced that the efficiency of Nanoblades can be further optimized through the use of modified sgRNAs, Cas9 bearing avidin sites to bind biotinilated donor DNA for efficient homologous recombination or even with the base editors recently developed by David R. Liu at Harvard but we believe that these are out of the scope of our current work.

In sum, this method is halfway between a viral and a non-viral approach. It is not likely to be as efficient as a viral approach, but one can make the reasonably argument that one is willing to sacrifice on efficiency in order to get other benefits associated with transient activity. In that case, however, the method had better compare well to the non-viral approaches that have this advantage. Since there is sparse comparison or benchmarking, it is not clear the method meets either bar.

Throughout our work, we have quantitatively shown that our VLP approach allows close to 100% of dsDNA breaks in U2OS cells, 85% in primary human fibroblasts, up to 67% of genome editing (assessed by throughput sequencing) in human iPScells and up to 76% in primary bone marrow cells (outperforming Cas9/sgRNA electroporation in this last cell type) in the absence of any selection step. We therefore do not think that our approach is less efficient than classical viral approaches and, to our knowledge, we are not aware of other delivery methods that can perform as well as Nanoblades in human iPS cells and primary bone marrow cells so rapidly and without a selection step. In most of the experiments that we present, we have monitored the efficiency of genome editing using validated tools (TIDE analysis, high-throughput sequencing, T7 endonuclease assays) so that readers can easily compare our results with those obtained by other groups with other methods. In the *in vivo* setups, we have now additional results (Figure 4e) showing that Nanoblades can achieve reasonable levels of genome editing in zygotes with mosaicism levels close to those reported by others (Chen et al 2016 and Yoon et al., 2018) using a simple and non-invasive injection strategy. Taken as a whole, our manuscript presents a wide range of possible applications for Nanoblades with efficiencies that are relevant.

We hope the reviewer will appreciate the extend of applications that are currently showcased in our manuscript, our efforts to characterize Nanoblades both in their composition and impact in target cells and the further developing potential to improve their usage.

Reviewers' Comments:

Reviewer #1:

Remarks to the Author:

The authors have complied with my previous requests but there is still one major point that needs to be clarified and a couple of observations/corrections.

The genetic background of mice in Fig. 4e needs to be clarified. The methods section states that the injected zygotes were obtained from crosses between C57Bl/6J mice (Non-agouti (a/a), black coat) and B6D2F1s. I assume that B6D2F1 are F1s from crosses between C57Bl/6J and DBA/2J mice (i.e. Jax stock 000671). DBA/2J mice have a complex coat color phenotype affected by multiple loci: Dilute (Myo5ad/Myo5ad), Brown (Tyrop1isa/Tyrop1isa) and Non-agouti (a/a). According to this genetic background, we should expect black (a/a), black/albino mosaics or albino (a/a; c/c) pups (if the wild-type albino allele C, was mutated by gene editing). However, the pictures in Fig.4e show agouti, agouti/albino mosaics or albino mice. This suggests that a wild-type allele of agouti (A) is segregating in these mice. Are the pups really derived from crosses between C57Bl/6J mice and B6D2F1s? If so, how can the coat color be explained? Also, does the Tyrosinase allele of DBA mice matches the guide RNAs used in this study?

Minor comments:

The albino mouse in figure 4e (#8) appears to be blind. This is interesting because DBA/2J mice have eye defects.

Fig.4c still refers to 2-cell embryos as "2 cells embryos" in grey boxes.

Reviewer #2:

Remarks to the Author:

The authors have conducted new experiments and introduced significant text revisions to address the prior reviews. As a result, they have addressed the concerns of this reviewer.

Point-by-point rebuttal:

Reviewer #1:

The authors have complied with my previous requests but there is still one major point that needs to be clarified and a couple of observations/corrections.

We thank the reviewer for her/his comments throughout the revision, which significantly improved the manuscript. In particular regarding transgenesis experiments.

The genetic background of mice in Fig. 4e needs to be clarified. The methods section states that the injected zygotes were obtained from crosses between C57Bl/6J mice (Non-agouti (a/a), black coat) and B6D2F1s. I assume that B6D2F1 are F1s from crosses between C57Bl/6J and DBA/2J mice (i.e. Jax stock 000671). DBA/2J mice have a complex coat color phenotype affected by multiple loci: Dilute (Myo5ad/Myo5ad), Brown (Tyrp1isa/Tyrp1isa) and Non-agouti (a/a). According to this genetic background, we should expect black (a/a), black/albino mosaics or albino (a/a; c/c) pups (if the wild-type albino allele C, was mutated by gene editing). However, the pictures in Fig.4e show agouti, agouti/albino mosaics or albino mice. This suggests that a wild-type allele of agouti (A) is segregating in these mice. Are the pups really derived from crosses between C57Bl/6J mice and B6D2F1s? If so, how can the coat color be explained? Also, does the Tyrosinase allele of DBA mice matches the guide RNAs used in this study?

We thank the reviewer for her/his noticing this discrepancy. Indeed, after checking with our animal facility, the pups derive from crosses between FVB/NRj females and B6D2F1 males and not from crosses between C57Bl/6J and DBA/2J mice as indicated in the methods section thus explaining the phenotype of the obtained F0 mice. We apologize for this mistake, we have now corrected the methods section accordingly. We have also checked the sequence of the Tyrosinase alleles from FVB.Nrj and B6D2F1 mice and all of them match the guide RNAs used in our study. We also confirm (based on the results of Sanger sequencing) that both alleles (maternal and paternal) were indeed edited in the genotyped mice.

Minor comments:

The albino mouse in figure 4e (#8) appears to be blind. This is interesting because DBA/2J mice have eye defects.

We checked with our animal facility and none of the mice have eye defects, the mouse must have had its eyes closed when taking the photograph.

Fig.4c still refers to 2-cell embryos as "2 cells embryos" in grey boxes.

We apologize for this oversight, we have now corrected the figure.

Reviewer #2 (Remarks to the Author):

The authors have conducted new experiments and introduced significant text revisions to address the prior reviews. As a result, they have addressed the concerns of this reviewer.

We thank the reviewer for her/his comments throughout the revision of the manuscript. We are glad that the new experiments addressed her/his concerns.